# Market framing bias and cross-sectional stock returns

**Jun Xie[1], Baohua Zhang[1], Bin Gao[2]***

**1** School of Economics, Guangxi University, Nanning, China, **2** School of Economics, Guangxi University for Nationalities, Nanning, China

\* financestudy@foxmail.com

**Data Availability Statement:** All relevant data are within the paper and its Supporting information files.

**Funding:** No. The authors would like to acknowledge the financial support from the National Natural Science Foundation of China

## Abstract

This paper introduces the market framing bias (MFB): a framing effect that affects the return-risk tradeoff under different frameworks of aggregate market losses and profits, which is measured by the absolute difference between betas in the rising and falling markets. The paper finds that the MFB can predict lower future stock return on the cross-section. Specifically, after controlling for various firm-specific characteristics, this predictive power of the FMB declines over time. Furthermore, the predictive power of the FMB is stable in the short term even after controlling for various pricing factors and firm-specific characteristics.

## Introduction

The framing effect is an anomaly that extensionally equivalent descriptions lead to different choices by altering the relative salience of different aspects of the problem [1]. Scholars have provided various pieces of evidence to reveal the framing effect in a stock market since the seminal work of Tversky and Kahneman [2], Steul [3], Grosshans and Zeisberger [4] and Bao et al. [5] argue that the framing effect influences investors. In fact, framing effects may result in multiple biases, for instance, narrow framing bias [6–8]: when thinking about a stock, they evaluate the return distribution of the stock itself; more sophisticated investors would evaluate the return distribution of the overall portfolio that results from tilting toward the stock. Gain/loss framing: Barbarab et al. [9] contend that position matters more in gain situations than in symmetric loss situations, in the specific case of gain/loss framing. However, these studies still only examine the presence or effects of the framing effect and do not measure it, leaving the question of how to measure the framing effect of stocks through publicly available trading data unanswered.

Unlike the previously mentioned papers, our analysis starts with the concept of market framing bias (MFB) which is the bias of the risk-return trade-off between the up and down markets. Indeed, the framing effect in behavioral finance is a cognitive bias where people make decisions based on whether they are in a loss-making or profit-making environment. The up or down markets are the natural positive framing (profit-making environment) or negative framing (loss-making environment) in the stock market. Thus, this bias of the risk-return trade-off between up and down markets is distinctly inconsistent with traditional CAPM [10, 11]. And

(No.72061002), the Natural Science Foundation of Guangxi Province of China (AD20159052), and the 'Guangxi One Thousand Young and Middle-Aged College and University Backbone Teachers Cultivation Program' Humanities and social sciences projects (2020QGRW016 and 2021QGRW003). The funders had no role in study design, data collection and analysis, decision to publish, or preparation of the manuscript.

**Competing interests:** The authors have declared that no competing interests exist.

it's also clearly different from narrow framing bias [7, 8], as the bias is mainly caused by the different market frameworks (so we call it market framing bias). In fact, the papers by Ang et al. [12] and Levi and Welch [13] explore the impact of downside beta on future returns from the perspective of risk (market risk exposures are time-vary). This paper, on the other hand, is the first to explore the significance of the difference between upside and downside beta from the perspective of investor psychology and behavioral biases, which is the biggest innovation of this paper. For example, Delta Beta is defined as the difference between downside beta and upside beta [13], which can be understood as the difference between downside exposure and upside exposure. But if you take an absolute value for Delta Beta, it's hard to explain from a risk perspective. However, the definition of frame bias is as long as there is "deviation", it does not involve "direction". Therefore, the absolute value of Delta Beta is interpreted as the bias of the risk-return trade-off between the up and down markets (MFB), which is a better operation than the risk perspective and is how we will calculate MFB in the next section.

Our paper then shifts to exploring whether MFB can predict the future returns of stocks in the cross-section. Based on the fact that investor irrationality can be translated into abnormal profits [14], we hypothesize that going long a portfolio of stocks with a low MFB and short-selling a portfolio of the stocks with a high MFB will earn abnormal returns over the next future months. Our paper thoroughly tested and proved this hypothesis through univariate and bivariate portfolio analysis, Fama-MacBeth regression [15], and out-of-sample analysis. We found that MFB can indeed predict future stock returns, which differs from Levi and Welch's analysis of Absolute Delta Beta. The primary reason for this difference is that our paper defines MFB based on short-term investor behavioral biases and calculates downside/upside betas using one month of data, while Levi and Welch [13], like Ang et al. [12], define downside/upside betas from the perspective of risk (market risk exposures are time-vary) and use one year of data to calculate them.

Our paper contributes three fresh insights into the framing effect measured by MFB. First, unlike the narrow framework [6–8], we give a measure of the framing effect in the market framework, which is noted as the market framing bias (MFB). The MFB defined in the paper may measure the framing effect under the different market frames, which is in line with the finding of Glascock and Lu-Andrews [16] that the beta coefficients of the market are different under different market states because of the framing effect. Second, the MFB can be used to predict future cross-sectional returns. The abnormal return produced by the MFB implies that the risk measured by MFB is not priced by various asset pricing factors, and it also illustrates that MFB's ability to predict future stock returns is different from that of the other firm-specific characteristics (Beta, Size, BM, STR, MoM, Illiq, Coskew, BD, VOLDU and VaR, detailed definitions of these firm-specific characteristics can be found in Section 3 of this paper.). Finally, we find an interesting but unproven phenomenon that, after controlling for various firm-specific characteristics, the predictive power of MFB for future stock returns decreases with increasing time to prediction.

The rest of the paper is organized as follows: Section 2 estimates the MFB. Section 3 describes the data and variables. Section 4 offers the predictive power of MFB for cross-sectional stock returns. Section 5 presents additional robustness tests. Section 6 concludes the study.

## Quantification of MFB

Current studies typically discuss the framing effect through experiments and surveys. However, the framing effect is not a laboratory curiosity, but a ubiquitous reality [1], and, to the best of our knowledge, this is the first time that the framing effect is measured under different

market frameworks, which is defined as the market framing bias (MFB): the bias of the risk-return trade-off between the up and down markets. Furthermore, we attempt to measure the MFB in terms of the absolute difference between the betas in the up and down markets, which is attributed to the following reasons:

First, the up or down market provides a "natural frame" to characterize the investment climate. Kahneman [1] points out that the framing effect is the anomaly that extensionally equivalent descriptions lead to different choices by altering the relative salience of different aspects of the problem (different environments). For investors, as observed by Glaser et al. [17], a positive frame (profit-making environment) or a negative frame (loss-making environment) may naturally indicate the up or down market. In other words, our definition of such a bias as MFB is similar to Kahneman's [1] definition of framing bias, except that our bias is produced under a different market framework. That's why we call it MFB, instead of just calling it framing bias.

Second, we use the difference between the betas in the up and down markets to measure the behavioral bias of investors. Glascock and Lu-Andrews [16] found that the market beta coefficients changed under different market states, which could be attributed to the framing effect. This evidence reveals that the framing effect in a stock market can be reflected by the difference between beta in different market situations (up or down markets), which inspires us to use the difference between betas in the up and down markets to measure the framing effect.

Finally, we take the absolute value, because the framing effect, as defined by Kahneman [1], has no positive or negative direction. It occurs when different descriptions of the framework lead to irrational decisions of investors. According to CAPM theory [10, 11], rational investors should have the same beta in both up and down markets. If the beta values differ across market status (frameworks), it reflects the existence of a framing effect for investors.

Thus, we define the absolute difference between betas in up and down markets as market framing bias (MFB), which measures a framing effect that influences people's decisions depending on whether they are in a loss-making or profit-making environment. Specifically, we denote δ as a binary variable that indicates up or down markets:

$$\delta = \begin{cases} 1, & \text{when the market premium is greater than } 0. \\ 0, & \text{when the market premium is lesser than or equal to } 0. \end{cases}$$

Following Glascock and Lu-Andrews [16], we estimate the difference between betas in up and down markets in the month $K$ by the following model:

$$R_{i,t} - R_{f,t} = \alpha_i + \beta_{i,K} \cdot (R_{M,t} - R_{f,t}) + \tilde{\beta}_{i,K} \cdot \delta_t \cdot (R_{M,t} - R_{f,t}) + \varepsilon_t. \tag{1}$$

Where $R_{i,t}$ is the return of stock $i$ at day $t$ in month $K$, $R_{f,t}$ is the risk-free interest rate at day $t$, $R_{M,t}$ is the return of the whole stock market on day $t$, $R_{M,t} - R_{f,t}$ is the market premium (market factor of CAPM), $\alpha_i$ is the intercept, $\beta$ and $\tilde{\beta}$ the regression coefficient, $\varepsilon_t$ is the residual. Obviously, if $\tilde{\beta}_{i,K} = 0$ then the model (1) turns into CAPM, which suggests the investors are rational. If $\tilde{\beta}_{i,K}$ is significantly different from zero, then $\tilde{\beta}_{i,K}$ is the difference between betas in up and down markets, since $(\beta_{i,K} + \tilde{\beta}_{i,K})$ is the beta in up markets and $\beta_{i,k}$ is the beta in down markets. Thus, we obtain market framing bias ($MFB_{i,k}$) of stock $i$ in month $K$ as

$$MFB_{i,K} = |\tilde{\beta}_{i,K}|. \tag{2}$$

The definition of MFB (MFB) in formula (2) follows Levi and Welch [13], except the following two aspects: Firstly, the reference points of the up and down markets are different. We divide the market into the up and down markets based on the reference point of zero rate of

return in formula (2), while Levi and Welch [13] use the historical average to divide the market. In fact, Ang et al. [12] construct the difference of downside/upside beta (Delta Beta) by considering the reference points of downward/upward market defined relative to the average market returns, riskless rate, and zero rate of return. We will discuss in detail the reference points in the robustness test. Secondly, the data used to calculate the upside and downside beta is different. We are based on short-term investor bias (framing bias) and use one month's data. However, similar to Ang et al. [12], Levi and Welch [13] adopted a one-year data perspective based on market risk exposures are time-vary. More importantly, we give a theoretical explanation of behavioral finance by linking the absolute difference between betas in up and down markets to the framing effect, whereas Levi and Welch [13] simply use it as a replacement indicator for an asymmetric beta. And acute problems encountered by Levi and Welch [13] in their US database are the stationarity and autocorrelation for the asymmetric beta, which do not affect us much, as this paper focuses on cross-sectional returns.

We conclude by noting that the MFB reflects how the risk-return trade-off varies between the up and down markets, similar to how investor expectations differ in the gain/loss framework. If the investor is rational for a stock $i$, then according to CAPM theory, the risk-return trade-offs should be the same under gain/loss framing, that is, $MFB_i = 0$. The larger the $MFB_i$, the higher level of investors' framing effect on the stock $i$.

The MFB is a cognitive bias, and the simultaneous existence of investors' rationality and cognitive biases thereby making investors adapt to the changing environment [18], which implies that the MFB may convey information that can forecast future stock returns. Thus, in the next sections, we will argue that MFB is a firm-specific characteristic that is useful for predicting future stock returns, but it is different from other firm-specific features and cannot be explained by various pricing factors.

## Data and variables

We collect the sample data for all A-shares (traded in the Shanghai Stock Exchange and Shenzhen Stock Exchange, excluding SSE STAR Market, as the official opening of the SSE STAR Market is July 22, 2019, which results in too little available data.). Daily and monthly stock market data used in this paper are from the RESSET database, except for the following data that come from China Stock Market & Accounting Research Database (CSMAR): momentum (UMD; Carhart, [19]), sentiment (SENT; Baker and Wurgler, [20]), monthly excess returns on the market (MKT), size (SMB), value (HML), investment (CMA) and profitability (RMW) factors of Fama and French [21]. The sample period is from January 2000 to December 2019, and stocks must have been traded for at least 36 months during the sample period. The final sample contains 3804 stocks and a total of nearly 500,000 firm-month observations.

We aim to analyze the role of MFB in predicting cross-sectional returns. Thus, we control for various firm-specific characteristics that affect expected stock returns. Specifically, the firm-specific characteristics are defined as follows. 1) Beta, following Bali et al. [22], the market beta of each stock with respect to the value-weighted market excess return calculated from daily returns during the month. 2) Size, coming from Fama and French [23], is calculated by the natural logarithm of each stock's market capitalization at the end of each month. 3) BM, book-to-market equity ratio at the end of each month, which also comes from Fama and French [23]. 4) STR, a short-term reversal, derived from Jegadeesh [24], is the return of a stock in the previous month. 5) MOM, the momentum return of each stock derived from Jegadeesh and Titman [25] is the cumulative return during the past 11 months after skipping one month. 6) Illiq, illiquidity coming from Amihud [26] is the absolute daily return divided by daily trading volume (hundred million yuan) averaged over all trading days in each month. 7) Coskew,

the co-skewness shown by Harvey and Siddique [27] is calculated as a daily regression coefficient $\beta_{2i}$ for the model $R_{i,t} - R_{f,t} = \alpha_i + \beta_{1i}MKT_t + \beta_{2i}MKT_t^2 + \varepsilon_t$ in each month. 8) BD, the downside beta shown by Ang et al. [12] and Chiang [28], is the sensitivity of each stock toward the excess market return during the days when the excess market return is below its mean during the month. 9) VOLDU, the difference between monthly money volume and its past 12-month average, which is derived from Atilgan et al. [29]. 10) VaR, value-at-risk also derived from Atilgan et al. [29], is calculated as the $1^{st}$ percentile of daily returns over the past 250 trading days at the end of the month.

The descriptive statistics and correlations with relative firm-specific characteristics used in this study are presented in Table 1, where the statistics are computed as time-series averages of the monthly cross-sectional means. Panel A of Table 1 shows that MFB has a mean equal to 1.12, a median equal to 1.00, and a standard deviation equal to 0.49. It appears that many stocks have quite respectable MFB spreads large enough to conclude that investors are always subject to framing effects and potentially allow for differential pricing effects of MFBs.

Panel B of Table 1 presents the time-series averages of cross-sectional correlations for all firm-specific characteristics, including the MFB. Generally, the MFB has no strong correlations with any of the firm-specific characteristics. Specifically, the correlation between co-skewness and MFB is -0.43, and the correlation between downside beta and MFB is 0.39, indicating that co-skewness and downside betas are weakly related to the MFB. Meanwhile, other

**Table 1. Descriptive statistics and correlation matrix of firm-specific variables.**

**Panel A: Descriptive statistics**

|  | MFB | Beta | Size | BM | STR | MOM | Illiq | Coskew | BD | VOLDU | VaR |
|---|---|---|---|---|---|---|---|---|---|---|---|
| Mean | 1.12 | 1.11 | 21.75 | 0.36 | 0.01 | 0.18 | 0.24 | -5.04 | 1.23 | 0.83 | 0.07 |
| St Dev | 0.49 | 0.18 | 0.99 | 0.13 | 0.10 | 0.52 | 0.28 | 11.09 | 0.34 | 15.65 | 0.01 |
| Median | 1.00 | 1.08 | 22.12 | 0.34 | 0.01 | 0.03 | 0.12 | -2.83 | 1.16 | -0.50 | 0.07 |
| Min | 0.29 | 0.59 | 19.75 | 0.17 | -0.29 | -0.60 | 0.01 | -88.98 | 0.25 | -49.90 | 0.05 |
| Max | 3.86 | 2.10 | 23.33 | 0.70 | 0.34 | 2.34 | 1.45 | 56.54 | 2.96 | 98.71 | 0.10 |
| Skew | 1.83 | 1.27 | -0.46 | 0.43 | 0.28 | 1.75 | 1.96 | -1.91 | 1.47 | 2.69 | 0.31 |
| Kurt | 6.20 | 4.94 | -1.18 | -0.75 | 1.08 | 3.45 | 3.63 | 19.04 | 5.49 | 14.88 | -0.68 |
| 10th Per | 0.63 | 0.95 | 20.29 | 0.20 | -0.09 | -0.29 | 0.04 | -14.89 | 0.95 | -10.92 | 0.06 |
| 90th Per | 1.72 | 1.34 | 22.93 | 0.55 | 0.14 | 0.80 | 0.61 | 1.60 | 1.61 | 12.22 | 0.10 |

**Panel B: Correlation matrix**

|  | MFB | Beta | Size | BM | STR | MOM | Illiq | Coskew | BD | VOLDU | VaR |
|---|---|---|---|---|---|---|---|---|---|---|---|
| MFB | 1.00 |  |  |  |  |  |  |  |  |  |  |
| Beta | 0.27 | 1.00 |  |  |  |  |  |  |  |  |  |
| Size | 0.29 | 0.27 | 1.00 |  |  |  |  |  |  |  |  |
| BM | 0.02 | 0.17 | -0.17 | 1.00 |  |  |  |  |  |  |  |
| STR | 0.10 | -0.09 | 0.12 | -0.21 | 1.00 |  |  |  |  |  |  |
| MOM | 0.01 | -0.17 | 0.30 | -0.55 | 0.37 | 1.00 |  |  |  |  |  |
| Illiq | -0.24 | -0.07 | -0.66 | 0.25 | -0.17 | -0.24 | 1.00 |  |  |  |  |
| Coskew | -0.43 | -0.31 | -0.20 | -0.04 | -0.01 | 0.01 | 0.16 | 1.00 |  |  |  |
| BD | 0.39 | 0.72 | 0.25 | 0.16 | -0.04 | -0.10 | -0.06 | -0.73 | 1.00 |  |  |
| VOLDU | 0.05 | -0.24 | 0.07 | -0.14 | 0.45 | 0.37 | 0.08 | 0.01 | -0.15 | 1.00 |  |
| VaR | -0.14 | 0.06 | 0.50 | -0.13 | 0.10 | 0.35 | -0.26 | 0.10 | 0.02 | -0.10 | 1.00 |

Note: Panel A reports the mean, standard deviation, median, minimum, maximum, skewness, kurtosis, 10th percentile, and 90th percentile for each variable. The statistics are calculated as the time-series averages of monthly cross-sectional means. Panel B reports the time-series average of monthly cross-sectional correlations among the variables. The sample period covers January 2000 to December 2019.

**Table 2. Stationary tests and correlation matrix of pricing factors.**

| | MKT | SMB | HML | RMW | CMA | UMD | SENT | PEAD | FIN | SMB$_Q$ | IA$_Q$ | ROE$_Q$ |
|---|---|---|---|---|---|---|---|---|---|---|---|---|
| MKT | 1.00 | | | | | | | | | | | |
| SMB | 0.08 | 1.00 | | | | | | | | | | |
| HML | -0.14 | -0.50 | 1.00 | | | | | | | | | |
| RMW | -0.28 | -0.78 | 0.27 | 1.00 | | | | | | | | |
| CMA | 0.14 | 0.42 | 0.14 | -0.67 | 1.00 | | | | | | | |
| UMD | -0.07 | -0.23 | -0.06 | 0.38 | -0.32 | 1.00 | | | | | | |
| SENT | -0.03 | 0.05 | -0.11 | -0.04 | 0.07 | 0.03 | 1.00 | | | | | |
| PEAD | -0.20 | -0.06 | 0.04 | 0.11 | -0.02 | -0.04 | -0.03 | 1.00 | | | | |
| FIN | -0.50 | -0.24 | 0.16 | 0.31 | -0.18 | -0.02 | 0.04 | 0.26 | 1.00 | | | |
| SMB$_Q$ | 0.04 | 0.92 | -0.43 | -0.66 | 0.35 | -0.20 | 0.05 | -0.07 | -0.17 | 1.00 | | |
| IA$_Q$ | -0.04 | 0.02 | 0.33 | -0.22 | 0.46 | -0.23 | -0.06 | -0.03 | 0.09 | 0.04 | 1.00 | |
| ROE$_Q$ | 0.20 | 0.63 | -0.12 | -0.80 | 0.64 | -0.52 | 0.04 | -0.01 | -0.20 | 0.48 | 0.16 | 1.00 |

Note: This table presents the correlation matrix for the monthly factors. The sample period is from January 2000 to December 2019.

firm-specific characteristics exhibit a weak correlation with the MFB, as indicated by the absolute values of all those correlation coefficients below 0.3. In addition, Panel B shows significant and negative correlations between size and illiquidity, co-skewness, and downside beta.

Since most asset pricing factors are derived from the monthly returns of a portfolio, they are likely to exhibit contemporaneous correlations. Table 2 presents the correlation matrix for monthly asset pricing factors (The asset pricing factors used in the paper include: FF5 factors (the market (MKT), size (SMB), value (HML), investment (CMA) and profitability (RMW) factors of Fama and French, [21]), Q4 factors (profitability (ROEQ), investment (IVAQ), market (MKT) and size (SMBQ) of Hou et al., [30]), SL2 factors (short-term behavioral factors (PEAD) and the long-behavioral factor (FIN) of Daniel et al., [31]), momentum (UMD; Carhart, [19]), and sentiment (SENT; Baker and Wurgler, [20]).). As expected, we find a correlation coefficient of 0.92 between SMB and SMB$_Q$, suggesting a strong positive correlation between the size factors. Therefore, when using a factor model that includes two size factors in our analysis, we avoid possible collinearity issues by using only the SMB factor in the following sections, although the collinearity issue has little impact on our conclusions.

## Cross-sectional return patterns associated with MFB

### Univariate and multi-term portfolio analysis

In this subsection, we use univariate portfolio-level analysis based on FF5 factors [21] to detect whether the MFB is a firm-special characteristic that can produce abnormal returns. We also conduct multi-term portfolio analysis is also implemented to investigate the multi-term predictive power of the MFB. And the multi-term excess return ($R_{i[t,t+\tau]}$) is the $\tau$-month ahead average excess return of stock $i$ for month $t$ to $t + \tau$ (in percentage), which is defined similarly to Birru and Young [32], but the average value is taken.

A univariate and multi-term portfolio-level analysis is performed where deciles are formed every month by ascending sorting stocks based on their MFB values. Excess returns and abnormal returns ($\alpha_{[t,t+\tau]}$, adjusted by the FF5 factors) from 1 to 12 months ahead are calculated for each decile to test whether the zero-cost portfolio (denoted as portfolio H-L), which takes a long position in stocks with the highest MFB and a short position in stocks with the lowest MFB, has a significant return.

**Table 3. Univariate and multi-term portfolio analysis.**

| | τ = 1 | τ = 2 | τ = 3 | τ = 4 | τ = 5 | τ = 6 | τ = 7 | τ = 8 | τ = 9 | τ = 10 | τ = 11 | τ = 12 |
|---|---|---|---|---|---|---|---|---|---|---|---|---|
| Panel A: Excess returns ($R_{[t,t+\tau]}$) | | | | | | | | | | | | |
| P1 | 0.76 | 0.53 | 0.56 | 0.58 | 0.53 | 0.40 | 0.44 | 0.54 | 0.54 | 0.45 | 0.45 | 0.53 |
| | (1) | (0.74) | (0.75) | (0.79) | (0.7) | (0.54) | (0.58) | (0.71) | (0.71) | (0.61) | (0.59) | (0.67) |
| P10 | -0.14 | 0.26 | 0.32 | 0.35 | 0.34 | 0.37 | 0.42 | 0.43 | 0.33 | 0.40 | 0.30 | 0.34 |
| | (-0.19) | (0.34) | (0.43) | (0.46) | (0.45) | (0.47) | (0.52) | (0.54) | (0.41) | (0.49) | (0.38) | (0.43) |
| H-L | -1.17‡ | -0.53‡ | -0.50‡ | -0.49‡ | -0.45‡ | -0.29† | -0.28† | -0.37‡ | -0.47‡ | -0.32† | -0.41‡ | -0.45‡ |
| | (-8.95) | (-3.86) | (-4.14) | (-3.79) | (-3.46) | (-2.10) | (-2.50) | (-3.04) | (-3.66) | (-2.32) | (-3.62) | (-3.14) |
| Panel B: Abnormal returns ($\alpha_{[t,t+\tau]}$) | | | | | | | | | | | | |
| P1 | -0.39‡ | -0.46‡ | -0.43‡ | -0.39‡ | -0.45‡ | -0.56‡ | -0.50‡ | -0.31† | -0.39‡ | -0.42‡ | -0.48‡ | -0.37‡ |
| | (-2.96) | (-3.62) | (-3.13) | (-2.88) | (-3.88) | (-3.96) | (-3.85) | (-2.20) | (-3.02) | (-3.81) | (-3.89) | (-2.73) |
| P10 | -1.18‡ | -0.77‡ | -0.70‡ | -0.66‡ | -0.66‡ | -0.66‡ | -0.54‡ | -0.61‡ | -0.58‡ | -0.57‡ | -0.67‡ | -0.58‡ |
| | (-7.52) | (-4.93) | (-5.11) | (-4.41) | (-4.70) | (-4.57) | (-3.69) | (-4.18) | (-4.30) | (-4.11) | (-4.4) | (-4.17) |
| H-L | -1.05‡ | -0.57‡ | -0.54‡ | -0.53‡ | -0.48‡ | -0.37‡ | -0.30‡ | -0.57‡ | -0.46‡ | -0.42‡ | -0.46‡ | -0.48‡ |
| | (-8.49) | (-4.25) | (-4.69) | (-4.74) | (-4.59) | (-3.08) | (-2.73) | (-4.23) | (-4.01) | (-3.95) | (-4.66) | (-3.95) |

Note: Panel A reports value-weighted excess returns (in percentage) from 1 to 12 months ahead after portfolio formation. Panel B presents the FF5-adjusted alphas for each decile from 1 to 12 months after portfolio formation. Newey and West [33] t-statistics are given in parentheses.

‡: Significant at 1%.

†: Significant at 5%.

*: Significant at 10%.

Table 3 presents the time-series averages of excess returns and alphas calculated after adjusting by the FF5 factors (MKT, SMB, HML, BMW, and CMA; Fama and French, [21]) for the MFB-sorted deciles. Panel A presents the value-weighted portfolio returns (in percentage) from 1 to 12 months after portfolio formation. We find that the excess returns of the lowest MFB deciles (P1) are always greater than those of the highest MFB deciles (P10). For instance, 12 months after portfolio formation, the excess return of the lowest MFB decile is 0.53%, whereas the excess return of the highest MFB decile is only 0.34%. Moreover, the differences in excess returns between the highest and lowest deciles (H-L) are all significantly negative after portfolio formation. For example, one month after the portfolio is formed, the excess return of portfolio H-L is -1.17% with a significant Newey and West [33] t-statistic of -8.95. Twelve months after the portfolio is formed, the excess return of portfolio H-L is -0.45% with a significant Newey and West [33] t-statistic of -3.14. The results are consistent with Breuer and Soypak [34], who emphasize that a stronger framing effect leads to a stronger status quo bias and negative outcomes.

Panel B of Table 3 examines whether the FF5 factor can explain the difference in excess returns between the extreme MFB deciles. In general, the abnormal returns (alphas) of the lowest MFB deciles are always greater than the alphas of the highest MFB deciles. For example, 1 month after portfolio formation, the alpha of the lowest and highest MFB deciles are -0.39% and -1.18%, respectively. And the alpha of the portfolio H-L is -1.05% with a significant Newey and West [33] t-statistic of -8.49. Thus, the results show that the predictive power of the MFB is not explained by the FF5 factors. Our results differ from Ang et al. [12]'s construction of Delta Beta (the difference between down beta and up beta) from the perspective of risk, for the following two main reasons: Firstly, we take the absolute value of Delta Beta from the perspective of investors' short-term behavioral biases, while it is hard to explain why Ang et al. [12] take absolute values from the perspective of risk. Secondly, when calculating the upside/

downside betas, we used daily data within each month based on investors' short-term behavioral biases, whereas Ang et al. [12] used daily data from the past year.

## Bivariate portfolio analysis

The poor explanation of the FF5 factors for the abnormal returns produced by the MFB presented in Table 3 is observed, possibly for two reasons. One reason is that a firm-specific characteristic correlated with MFB but not captured by the FF5 factors has a significant impact on expected stock returns. Another reason is that the market may be influenced by rational and irrational forces [5], and only using the FF5 factors in the above analysis may be insufficient. Thus, bivariate portfolio analysis is used here to test whether there are other firm-specific characteristics (see section 3 for detailed definitions) and behavior-related asset pricing factors (for example, SL2, UMD, and SENT factors) that are correlated with the MFB.

Bivariate two-stage 10×10 dependent sorts are used to do bivariate portfolio analysis. First, we sort stocks into decile portfolios monthly based on various firm-specific characteristics (Beta, Size, BM, STR, MoM, Illiq, Coskew, BD, VOLDU, and VaR). Second, we sort stocks into additional deciles based on the MFB within each firm-specific characteristic decile that is sorted in step one. Then, 100 conditionally double-sorted groups are provided to construct ten portfolios through the above two steps. Portfolio 1 is the combined group of stocks with the lowest MFB in each firm-specific characteristic decile, whereas portfolio 10 is the one with the highest MFB.

Table 4 presents the results of the bivariate portfolio analysis (for brevity, only the results of 1, 3, 6, and 12 months after portfolio formation are reported). Panel A shows the excess returns of the portfolio H-L by controlling for various firm-special characteristics. It is easy to find that all excess returns are significantly smaller than zero, except when controlled by BM ($\tau = 12$), MoM ($\tau = 12$) and VOLDU ($\tau = 6, 12$), respectively. These results simply show that the role of MFB on future stock returns may be consistent with BM, MoM, and VOLDU in the long run, but BMF is not replaced by these firm-special characteristics because the short-term role cannot be replaced by other firm characteristics. Furthermore, we can find that, after controlling for these firm-special characteristics, MFB's fine predictive power diminishes over time. For example, when Beta is the first-stage sorting variable, the excess return of the portfolio H-L changes from -0.89% at $\tau = 1$ to -0.15% at $\tau = 12$.

Panel B shows the abnormal returns, adjusted by FF5 factors, for the portfolio H-L grouped by the MFB at the second-stage. First, all abnormal returns adjusted by FF5 factors are significantly negative in the short-term ($\tau = 1, 3$), which implies that FF5 factors cannot explain the excess returns produced by the bivariate sorts in the short-term. Second, consistent with panel A, we also find that the ability of MFB to generate abnormal returns still fades over time after controlling for these firm-special characteristics. For example, when Size is the first-stage sorting variable, the abnormal returns are -0.86%, -0.46%, -0.34%, and -0.22% at t+1, t+3, t+6, and t+12, respectively.

Concerning the possibility that the FF5 factors are not sufficient to explain the excess returns generated by the FMB, in Panel C we include more factors to explore this issue. Panel C in Table 4 presents the results adjusted by more asset pricing factors, which include FF5 factors (MKT, SMB, HML, RMW, and CMA; Fama and French, [21]), SL2 factors (PEAD, short-term behavioral factors, and FIN, the long-term behavioral factor; Daniel et al., [30]), momentum (UMD; Carhart, [19]), sentiment (SENT; Baker and Wurgler, [20]). Clearly, all abnormal returns are significantly negative in the short term, which means the addition of more pricing factors does not eliminate the predictive power of MFB.

**Table 4. Bivariate portfolio analysis.**

| | Beta | Size | BM | STR | MoM | Illiq | Coskew | BD | VOLDU | VaR |
|---|---|---|---|---|---|---|---|---|---|---|
| Panel A: Return of H-L | | | | | | | | | | |
| τ = 1 | -0.89‡ | -0.86‡ | -0.76‡ | -0.88‡ | -0.89‡ | -0.89‡ | -0.50‡ | -0.76‡ | -0.49‡ | -0.76‡ |
| | (-6.92) | (-6.47) | (-6.28) | (-7.22) | (-7.13) | (-6.82) | (-4.98) | (-6.76) | (-3.53) | (-5.98) |
| τ = 3 | -0.41‡ | -0.46‡ | -0.31‡ | -0.43‡ | -0.39‡ | -0.40‡ | -0.29‡ | -0.31‡ | -0.20* | -0.36‡ |
| | (-4.06) | (-4.61) | (-3.07) | (-4.32) | (-4.04) | (-3.62) | (-4.34) | (-4.09) | (-1.7) | (-4.47) |
| τ = 6 | -0.23‡ | -0.34‡ | -0.14* | -0.23† | -0.21‡ | -0.27‡ | -0.21‡ | -0.16‡ | -0.12 | -0.19‡ |
| | (-2.78) | (-4.74) | (-1.68) | (-2.42) | (-2.7) | (-2.95) | (-4.71) | (-2.61) | (-1.24) | (-2.85) |
| τ = 12 | -0.15† | -0.22‡ | -0.05 | -0.16† | -0.10 | -0.18‡ | -0.12‡ | -0.11† | -0.06 | -0.13† |
| | (-2.36) | (-4.09) | (-0.78) | (-2.13) | (-1.41) | (-2.64) | (-3.21) | (-2.48) | (-0.82) | (-2.34) |
| Panel B: Abnormal return adjusted by FF5 | | | | | | | | | | |
| τ = 1 | -0.77‡ | -0.68‡ | -0.63‡ | -0.77‡ | -0.79‡ | -0.74‡ | -0.47‡ | -0.62‡ | -0.37‡ | -0.68‡ |
| | (-6.99) | (-5.73) | (-5.62) | (-6.79) | (-7.45) | (-6.38) | (-4.37) | (-5.97) | (-3.42) | (-5.29) |
| τ = 3 | -0.51‡ | -0.55‡ | -0.41‡ | -0.53‡ | -0.48‡ | -0.49‡ | -0.34‡ | -0.39‡ | -0.31‡ | -0.45‡ |
| | (-6.00) | (-6.80) | (-4.93) | (-6.32) | (-5.75) | (-5.42) | (-4.30) | (-5.39) | (-3.30) | (-5.38) |
| τ = 6 | -0.29‡ | -0.37‡ | -0.21‡ | -0.29‡ | -0.27‡ | -0.32‡ | -0.21‡ | -0.18‡ | -0.16† | -0.23‡ |
| | (-4.78) | (-7.03) | (-3.94) | (-4.38) | (-4.62) | (-4.66) | (-4.53) | (-3.51) | (-2.25) | (-4.70) |
| τ = 12 | -0.18‡ | -0.23‡ | -0.08 | -0.18‡ | -0.11* | -0.20‡ | -0.12‡ | -0.12‡ | -0.08 | -0.13‡ |
| | (-2.98) | (-4.63) | (-1.28) | (-2.71) | (-1.81) | (-3.29) | (-2.89) | (-2.86) | (-1.22) | (-2.62) |
| Panel C: Abnormal return adjusted by FF5+SL2+UMD+SENT | | | | | | | | | | |
| τ = 1 | -0.93‡ | -0.86‡ | -0.76‡ | -0.93‡ | -0.91‡ | -0.92‡ | -0.54‡ | -0.75‡ | -0.47‡ | -0.86‡ |
| | (-7.85) | (-6.69) | (-6.59) | (-7.28) | (-8.12) | (-7.15) | (-4.46) | (-6.30) | (-3.93) | (-6.40) |
| τ = 3 | -0.58‡ | -0.61‡ | -0.45‡ | -0.60‡ | -0.53‡ | -0.56‡ | -0.39‡ | -0.41‡ | -0.35‡ | -0.48‡ |
| | (-5.48) | (-6.07) | (-4.28) | (-5.60) | (-5.36) | (-4.81) | (-4.35) | (-4.85) | (-3.05) | (-5.17) |
| τ = 6 | -0.28‡ | -0.36‡ | -0.21‡ | -0.29‡ | -0.29‡ | -0.31‡ | -0.19‡ | -0.14† | -0.16* | -0.22‡ |
| | (-4.14) | (-5.98) | (-3.15) | (-3.59) | (-4.31) | (-4.04) | (-3.42) | (-2.41) | (-1.90) | (-3.86) |
| τ = 12 | -0.17† | -0.22‡ | -0.07 | -0.17† | -0.11 | -0.21‡ | -0.10† | -0.10† | -0.07 | -0.11† |
| | (-2.57) | (-3.98) | (-1.04) | (-2.34) | (-1.60) | (-3.13) | (-2.32) | (-2.21) | (-1.02) | (-2.03) |

**Note:** Reported are the results of value-weighted bivariate portfolio analysis for future 1, 3, 6, and 12 months (τ = 1, τ = 3, τ = 6, and τ = 12). Only the results for the portfolio H-L are reported in the table. Newey and West [33] t-statistics are given in parentheses.

‡: Significant at 1%.

†: Significant at 5%.

*: Significant at 10%.

In summary, by the bivariate portfolio analysis, we show that a higher FMB still has a lower future return and vice versa, especially in the short term. Moreover, this negative predictive power of the FMB weakens over time, but cannot fade by controlling for various firm-specific characteristics and asset pricing factors.

## Firm-level Fama-MacBeth regressions

In this subsection, firm-level Fama-MacBeth regressions (Fama-MacBeth, [15]) are used to examine the negative predictive power of MFB for future stock returns, by controlling for the other 10 firm-specific characteristics (Beta, Size, BM, STR, MoM, Illiq, Coskew, BD, VOLDU and VaR) that may determine future returns.

In the first stage, monthly cross-sectional regressions of excess stock returns ($R_{i[t+1,t+\tau]}$) on the values of the MFB and aforementioned ten control variables are measured in month $t$.

**Table 5. Firm-level Fama-MacBeth regression.**

| | $\tau = 1$ | | $\tau = 3$ | | $\tau = 6$ | | $\tau = 12$ | |
|---|---|---|---|---|---|---|---|---|
| | (1) | (2) | (1) | (2) | (1) | (2) | (1) | (2) |
| MFB | -2.23‡ | -2.22‡ | -2.21‡ | -1.90‡ | -1.91‡ | -1.65‡ | -1.58‡ | -1.04‡ |
| | (-6.81) | (-5.59) | (-5.59) | (-4.87) | (-4.12) | (-4.01) | (-3.17) | (-2.67) |
| Beta | | -3.84‡ | | -6.33‡ | | -9.33‡ | | -12.67‡ |
| | | (-3.78) | | (-3.58) | | (-4.03) | | (-4.53) |
| Size | | 1.40† | | 1.73* | | 1.60 | | 1.75 |
| | | (2.28) | | (1.78) | | (1.37) | | (1.34) |
| BM | | -0.67 | | -0.22 | | 0.41 | | 0.66 |
| | | (-1.43) | | (-0.51) | | (0.98) | | (1.61) |
| STR | | 1.90‡ | | 1.87* | | 0.66 | | -1.34 |
| | | (2.77) | | (1.88) | | (0.56) | | (-1.17) |
| MoM | | 4.83 | | 9.91† | | 9.38* | | 10.45† |
| | | (1.60) | | (2.00) | | (1.95) | | (2.05) |
| Illiq | | 0.44 | | 1.53* | | 1.26 | | 0.77 |
| | | (0.51) | | (1.72) | | (1.23) | | (1.02) |
| Coskew | | 0.18 | | 1.46 | | 1.37 | | 0.86 |
| | | (0.22) | | (1.64) | | (1.20) | | (1.17) |
| Betadown | | -3.37‡ | | -3.22‡ | | -2.86‡ | | -2.17‡ |
| | | (-8.04) | | (-6.32) | | (-5.06) | | (-3.62) |
| Voldu | | -2.86‡ | | -3.31‡ | | -3.49‡ | | -3.01‡ |
| | | (-7.23) | | (-5.32) | | (-4.30) | | (-3.30) |
| VaR | | 3.24‡ | | 1.16 | | -0.57 | | -1.55* |
| | | (4.78) | | (1.57) | | (-0.68) | | (-1.69) |
| Adj. $R^2$ | 0.47 | 9.19 | 0.30 | 8.61 | 0.22 | 8.49 | 0.18 | 9.45 |

Note: This table reports the results of the Fama-MacBeth cross-sectional regressions of individual firms for future 1, 3, 6, and 12 months ($\tau = 1$, $\tau = 3$, $\tau = 6$, and $\tau = 12$) on the control variables measured in month $t$. The column labeled "(1)" or "(2)" presents the average coefficient and the adjusted $R^2$ (in percentage) of the univariate regression or the full regression specification, respectively. Newey and West [33] t-statistics are given in parentheses.

‡: Significant at 1%.

†: Significant at 5%.

*: Significant at 10%.

Specifically, the cross-sectional model estimated monthly is

$$R_{i,[t,t+\tau]} = \alpha_{t+1} + \beta_{1,t+1}MFB_{i,t} + \beta_{2,t+1}Controls_{i,t} + \varepsilon_{i,t+1}. \tag{3}$$

The univariate regression or the full regression specification of the model (3) are estimated by using the ordinary least squares (OLS) methodology. Furthermore, to assess the economic significance, all variables in the model (3) are standardized.

In the second stage, the cross-sectional regression coefficients are estimated by the time-series averages where Newey and West [33] t-statistics are used. The coefficients of the model Table 5 and the average adjusted $R^2$ are reported in Table 5 for the univariate regression (column (1)) and the full regression (column (2)) specification, respectively.

In the short-term $\tau = 1$ case, the average slope coefficient from the univariate regression in column (1) is -2.23 with a t-statistic of -6.81, which implies that if MFB increases one standard deviation in month $t$, the one-month-ahead excess return will decrease -2.23% on average. The average slope coefficient from the full regression in column (2) is -2.22 with a t-statistic of -5.59, which implies that if the MFB increases one standard deviation in month $t$, the one-

month-ahead excess return will decrease -2.22% on average, after controlling for other ten firm-special characteristics. When $\tau = 3$, $\tau = 6$, and $\tau = 12$, the results do not differ much from the short-term $\tau = 1$, indicating that the other ten firm-special characteristics added as control variables in the full regression, do not significantly interfere with the predictive power of the MFB. In fact, we confirm again that the predictive power of the MFB is mainly weakening with time: from the univariate regressions, an increase of one standard deviation in the MFB in month $t$, will lead the average slope coefficient to degrade from -2.23 ($\tau = 1$) to -1.58 ($\tau = 12$), and the results from the full regressions are similar.

We also find that, from the full regression, the variables that have a stable effect on future returns, both in the short and long term, are Beta, BD, and Voldu. And in the short-term ($\tau = 1$), different from the results of Atilgan et. al [29], the significant coefficient estimates indicate that stocks with and higher size (Size), higher return in the previous month (STR), and higher left-tail risk (VAR) are associated with higher expected returns, which implies that MFB may have replaced their negative influence and caused new anomalies.

## Out-of-sample predictability

The in-sample tests in the above discussion may present a look-ahead bias. Following Goyal and Welch [35] and Chen et al. [36], out-of-sample prediction performance can help validate the in-sample performance. Thus, in this subsection, the predictive ability of MFB is evaluated based on out-of-sample tests.

Our test data are the value-weighted MFB-sorted quintile portfolios. At first, in the 100 months rolling (expanding) window, the $t + \tau$ month's return of a portfolio $p$ ($R_{p[t,t+\tau]}$) is forecasted by the following classic asset pricing factors model:

$$R_{p,[t,t+\tau]} = \alpha + \beta_Z \cdot Z_t + \varepsilon_{p,t+\tau}, \tag{4}$$

where $Z_t$ is the pricing factors FF5, Q4, and FF5+IVA$_Q$+ROE$_Q$+SL2+UMD+SENT (labeled as ALL in the table) in month $t$, respectively. Based on asset pricing theory, the non-intercept regression is adopted to estimate the model (4), since the $\alpha$ is zero in an efficient market. Next, we forecast the $t + \tau$ month's return of our model (8) that the MFB is added in the benchmark model (4) by non-intercept regression:

$$R_{p,[t,t+\tau]} = \alpha + \beta_p MFB_{p,t} + \beta_{p,Z} \cdot Z_t + \varepsilon_{p,t+\tau}, \tag{5}$$

where $MFB_{p,t}$ is the value-weighted MFB of the stocks in portfolio $p$ for month $t$. Third, based on the coefficient estimates from the benchmarking model (4) and our model (5), we compute the forecast of the equity premium for the next month ($R_{p[t,t+\tau+1]}$). Finally, the out-of-sample $R_{oos}^2$ is calculated following Campbell and Thompson [37] and Goyal and Welch [35]

$$R_{oos}^2 = 1 - \frac{\sum_{i=1}^{T-t-\tau} \left(R_{p,[t,t+\tau+i]} - \hat{R}_{p,[t,t+\tau+i]}\right)^2}{\sum_{i=1}^{T-t-\tau} \left(R_{p,[t,t+\tau+i]} - \tilde{R}_{p,[t,t+\tau+i]}\right)^2}, \tag{6}$$

where $R_{p[t,t+\tau+i]}$, $\hat{R}_{p,[t,t+\tau+i]}$ and $\tilde{R}_{p,[t,t+\tau+i]}$ is the true excess return of the portfolio $p$, the predicted excess return of our model (5), and the predicted excess return of the classic asset pricing model (4), respectively. Obviously, $R_{oos}^2 > 0$ implies that our model (5) has higher prediction accuracy than the benchmark model (4). In addition, different from traditional $R^2$ estimated from an in-sample test, out-of-sample $R_{oos}^2$ can be negative when the forecast error of our model (5) is higher than that of the benchmark model (4). Furthermore, following Clark and West [38], the MSFE-adjusted statistics to evaluate the statistical significance of $R_{oos}^2$ is

estimated as the t-statistic of regress $\tilde{d}_Z$ on a constant:

$$\tilde{d}_Z = u_Z^2 - \left[ u_{MFB}^2 - (\tilde{R} - \hat{R})^2 \right], \qquad (7)$$

where $u_Z$ is the forecast error for the benchmark model (4), $u_{MFB}$ is the forecast error for our forecasting model (5). The out-of-sample test results are reported in Table 6.

Panel A of Table 6 shows the 1-month ahead forecasting ability of MFB. When a 100-month rolling window is used, $R_{oos}^2$ are significantly positive, except that the lowest MFB quintile and FF5+IVA$_Q$+ROE$_Q$+SL2+UMD+SENT factors are used in the benchmarking model. When a 100-month expanding window is used, $R_{oos}^2$ are significantly positive, except that the $R_{oos}^2$ is -8.43 with a t-statistic of 1.94 for the lowest MFB quintile and using the FF5

**Table 6. Out-of-sample predictability.**

| | Rolling approach | | | | | Expanding approach | | | | |
|---|---|---|---|---|---|---|---|---|---|---|
| | MFB1 | MFB2 | MFB3 | MFB4 | MFB5 | MFB1 | MFB2 | MFB3 | MFB4 | MFB5 |
| Panel A: 1-month ahead ($\tau = 1$) | | | | | | | | | | |
| FF5 | 6.12‡ | 19.32‡ | 16.71‡ | 13.62‡ | 29.05‡ | 3.99‡ | 15.26‡ | 10.90‡ | 7.52† | 27.08‡ |
| | (4.27) | (6.50) | (4.99) | (3.57) | (6.94) | (4.46) | (6.27) | (4.79) | (2.57) | (7.13) |
| Q4 | 8.09‡ | 15.54‡ | 12.14‡ | 5.73† | 17.13‡ | 7.01‡ | 14.58‡ | 11.73‡ | 6.33† | 17.13‡ |
| | (2.99) | (3.53) | (3.21) | (2.40) | (4.31) | (3.30) | (3.87) | (3.47) | (2.22) | (4.82) |
| All | -7.34 | 10.05‡ | 9.74‡ | 9.36‡ | 23.37‡ | -8.43* | 5.37‡ | 1.98‡ | 3.08† | 21.38‡ |
| | (1.66) | (4.74) | (4.06) | (3.64) | (4.64) | (1.94) | (5.09) | (4.07) | (2.31) | (5.67) |
| Panel B: 3-month ahead ($\tau = 3$) | | | | | | | | | | |
| FF5 | 5.57‡ | 18.86‡ | 16.12‡ | 13.22‡ | 28.13‡ | 3.47‡ | 15.29‡ | 10.88‡ | 7.48† | 26.35‡ |
| | (4.15) | (6.41) | (4.89) | (3.46) | (6.70) | (4.32) | (6.19) | (4.70) | (2.51) | (6.87) |
| Q4 | 7.93‡ | 15.84‡ | 12.39‡ | 6.23† | 15.83‡ | 6.86‡ | 14.93‡ | 12.11‡ | 6.63† | 15.93‡ |
| | (2.94) | (3.50) | (3.18) | (2.40) | (4.10) | (3.22) | (3.83) | (3.44) | (2.23) | (4.58) |
| All | -8.12 | 9.66‡ | 9.16‡ | 9.14‡ | 22.46‡ | -9.40* | 5.08‡ | 1.61‡ | 2.93† | 20.62‡ |
| | (1.59) | (4.65) | (3.97) | (3.55) | (4.46) | (1.81) | (4.99) | (3.94) | (2.24) | (5.43) |
| Panel C: 6-month ahead ($\tau = 6$) | | | | | | | | | | |
| FF5 | 4.12‡ | 16.74‡ | 14.43‡ | 12.17‡ | 27.16‡ | 2.79‡ | 14.21‡ | 10.54‡ | 7.85† | 25.67‡ |
| | (3.95) | (6.13) | (4.67) | (3.28) | (6.56) | (4.17) | (6.00) | (4.60) | (2.52) | (6.72) |
| Q4 | 7.36‡ | 14.89‡ | 11.80‡ | 5.76† | 15.77‡ | 6.20‡ | 14.00‡ | 11.56‡ | 6.34† | 16.08‡ |
| | (2.84) | (3.37) | (3.09) | (2.29) | (4.06) | (3.08) | (3.67) | (3.33) | (2.15) | (4.56) |
| All | -9.80 | 7.35‡ | 7.59‡ | 8.29‡ | 21.81‡ | -10.46* | 3.76‡ | 1.17‡ | 3.17† | 20.04‡ |
| | (1.47) | (4.42) | (3.79) | (3.38) | (4.34) | (1.70) | (4.82) | (3.85) | (2.24) | (5.31) |
| Panel D: 12-month ahead ($\tau = 12$) | | | | | | | | | | |
| FF5 | 1.61‡ | 12.67‡ | 10.58‡ | 10.18‡ | 24.79‡ | 1.37‡ | 12.29‡ | 8.23‡ | 8.12† | 24.35‡ |
| | (3.58) | (5.50) | (4.12) | (2.83) | (6.16) | (3.88) | (5.56) | (4.20) | (2.40) | (6.39) |
| Q4 | 4.57† | 10.8‡ | 7.99† | 3.55* | 14.41‡ | 4.71‡ | 11.99‡ | 9.41‡ | 5.26* | 16.29‡ |
| | (2.46) | (2.96) | (2.65) | (1.82) | (3.87) | (2.75) | (3.29) | (2.94) | (1.84) | (4.44) |
| All | -15.27 | -1.37‡ | 0.26‡ | 6.22‡ | 16.69‡ | -12.11 | 1.46‡ | -1.22‡ | 3.61† | 18.29‡ |
| | (1.11) | (4.10) | (3.30) | (2.82) | (4.66) | (1.49) | (4.37) | (3.44) | (2.19) | (5.27) |

Note: We test the forecasting power of the MFB for the $\tau$-month-ahead excess return of the value-weighted MFB-sorted quintile portfolios ($\tau = 1$, $\tau = 3$, $\tau = 6$, and $\tau = 12$). The benchmark models are FF5, Q4, and the factors model, respectively. Panel A, B, C, and D show the 1, 3, 6, and 12-month ahead forecasting ability of MFB, respectively.

*p < .1;

†p < .05;

‡p < .01 (the significance levels for two-sided tests are indicated).

+IVA$_Q$+ROE$_Q$+SL2+UMD+SENT factors in the benchmarking model. The results indicate that, in most cases, the MFB has a predictive power that is different from the benchmark pricing factors to the 1-month ahead stock return.

Panel B and Panel C of Table 6 also have similar patterns to Panel A, and the $R^2_{oos}$ (absolute value) becomes smaller for the lowest MFB quintile and when using the FF5+IVA$_Q$+ROE$_Q$+SL2+UMD+SENT factors in the benchmarking model. In particular, when predicting the 12-month ahead stock returns in Panel D, there are two quintiles with negative $R^2_{oos}$ for either the rolling method or expanding method. For example, when the rolling method and the FF5+IVA$_Q$+ROE$_Q$+SL2+UMD+SENT factors are used, the out-of-sample $R^2_{oos}$ of the lowest two quintiles are -15.27 and -1.37, respectively. These results suggest that the predictive power of MFB for future returns is weakening as the prediction time increases.

Taken together, the results of the out-of-sample tests show that our predictive model with MFB outperforms the benchmark pricing model in predicting equity premiums, and in agreement with the in-sample analysis, the predictive power of MFB for future returns weakens as the prediction time increases.

## Robustness tests

The utilization of publicly traded data to define and quantify framework deviations for the first time is expected to generate considerable debate. Firstly, as we are aware, stocks tend to exhibit stronger co-movement during market downturns [39]. While our previous analysis may have overlooked this phenomenon, the collective effect of these investors will diminish the influence of framing bias on future stock returns. Furthermore, this consistent co-movement effect may result in downside betas having a tendency to be higher than upside betas, thereby potentially mitigating the significance of absolute value calculations of MFB. Lastly, does the choice of reference points for market up/down impact the measurement of MFB? In this section, we provide answers to the aforementioned questions using stability tests. And, for the purpose of a concise introduction, we shall solely examine the impact of MFB on the stock's returns within the upcoming month.

### Subsample analysis based on the boom and recession

In this section, we divide the full sample period into two based on several macroeconomic indicators that signal whether the economy is experiencing recessions or booms. To measure the state of China's economy, we choose the following three macroeconomic indicators from the National Bureau of Statistics: Purchasing manager's index (PMI), an important evaluation index of economic activities and a barometer of economic changes; Economic sentiment leading index (ESLI), which is composed of a set of leading indicators that lead the consensus index and is used to predict the future trend of the economy; Economic sentiment consistent index (ESCI), which reflects the basic trend of the current economy.

Our sample from January 2005 to December 2019 (Since China's PMI index started in 2005) is split into two subsamples by the three indicators, and univariate analysis is made for them. Specifically, after splitting the sample one at a time, quintiles are formed every month by sorting stocks based on ascending MFB, and the monthly value-weighted excess returns (ER) are calculated for each quintile. The alphas ($\alpha$) to all quintiles, including zero-investment portfolio (H-L) which is long (short) in equities with high (low) framing effect, are obtained by regression from FF5+IVA$_Q$+ROE$_Q$+SL2+UMD+SENT. Table 7 presents the results, where two subsamples are split by PMI, ESLI and ESCI respectively.

Panel A presents the results for PMI which is used to split the sample. When PMI>50, macroeconomic conditions are booming, the left columns of Panel A show that the excess return

**Table 7. Subsample analysis.**

Panel A: Subsample divided by PMI

| | | Booms: PMI>50 | | | | | Recessions: PMI< = 50 | | | | | |
|---|---|---|---|---|---|---|---|---|---|---|---|---|
| | Port 1 | Port 2 | Port 3 | Port 4 | Port 5 | H-L | Port 1 | Port 2 | Port 3 | Port 4 | Port 5 | H-L |
| ER | 1.32 | 1.25 | 1.51 | 1.05 | 0.70 | -0.90‡ | -0.90 | -1.75 | -1.20 | -1.40 | -1.50 | -0.87 |
| | (1.50) | (1.46) | (1.43) | (1.17) | (0.78) | (-3.16) | (-0.89) | (-1.36) | (-0.96) | (-1.17) | (-1.07) | (-1.61) |
| α | -0.19 | -0.08 | -0.10 | -0.49‡ | -1.10‡ | -1.20‡ | -0.05 | -0.57† | -0.08 | -0.88‡ | -1.15 | -1.36 |
| | (-1.02) | (-0.47) | (-0.45) | (-2.99) | (-4.58) | (-4.73) | (-0.17) | (-2.09) | (-0.27) | (-3.49) | (-1.47) | (-1.64) |

Panel B: Subsample divided by ESLI

| | | Booms: ESLI>100 | | | | | Recessions: ESLI< = 100 | | | | | |
|---|---|---|---|---|---|---|---|---|---|---|---|---|
| | Port 1 | Port 2 | Port 3 | Port 4 | Port 5 | H-L | Port 1 | Port 2 | Port 3 | Port 4 | Port 5 | H-L |
| ER | 0.87 | 0.70 | 1.04 | 0.42 | -0.07 | -1.21‡ | 1.02 | 0.72 | 0.98 | 1.03 | 1.15 | -0.19 |
| | (0.92) | (0.76) | (0.87) | (0.44) | (-0.07) | (-3.84) | (0.87) | (0.62) | (0.80) | (0.77) | (0.91) | (-0.41) |
| α | -0.14 | -0.15 | -0.07 | -0.60‡ | -1.05‡ | -1.19‡ | 0.05 | -0.41† | -0.12 | -0.24 | -0.91† | -1.27† |
| | (-0.80) | (-0.90) | (-0.28) | (-3.50) | (-3.94) | (-4.20) | (0.13) | (-2.30) | (-0.52) | (-0.94) | (-2.02) | (-2.42) |

Panel C: Subsample divided by ESCI

| | | Booms: ESCI>100 | | | | | Recessions: ESCI< = 100 | | | | | |
|---|---|---|---|---|---|---|---|---|---|---|---|---|
| | Port 1 | Port 2 | Port 3 | Port 4 | Port 5 | H-L | Port 1 | Port 2 | Port 3 | Port 4 | Port 5 | H-L |
| ER | 0.50 | 0.24 | 0.76 | -0.02 | -0.58 | -1.36‡ | 1.36 | 1.20 | 1.30 | 1.28 | 1.26 | -0.40 |
| | (0.42) | (0.20) | (0.48) | (-0.02) | (-0.47) | (-4.01) | (1.49) | (1.40) | (1.44) | (1.35) | (1.33) | (-1.28) |
| α | -0.25 | -0.13 | 0.12 | -0.48† | -1.33‡ | -1.37‡ | -0.04 | -0.28* | -0.34* | -0.56† | -0.85‡ | -1.09‡ |
| | (-1.27) | (-0.67) | (0.43) | (-2.45) | (-4.62) | (-4.58) | (-0.20) | (-1.8) | (-1.99) | (-2.57) | (-3.17) | (-3.11) |

Note: Newey and West [33] t-statistics are given in parentheses.

‡: Significant at 1%.

†: Significant at 5%.

*: Significant at 10%.

decreased from 0.5 for the lowest MFB quintile to -0.58 for the highest MFB quintile. And the excess return of the portfolio H-L is -0.90 with a t-statistic of -3.16. However, when PMI< = 50, macroeconomic conditions are recessions, the right columns of Panel A show that the excess return of zero-investment portfolio is -0.87% with an insignificant t-statistic of -1.61. The results imply that the predictive ability of MFB in the booms subsample (PMI>50) are significant, but not during market downturns (PMI< = 50).

This conclusion is confirmed by Panel B and Panel C in Table 7. For example, when the ESCI< = 100 and the market declines, the excess return of the portfolio H-L is -0.40 with a t-statistic of -1.28 (At this time, although alpha is -1.09 with a t-statistic of -3.11, it only shows that FF5, Q4, SL2, momentum, and investor sentiment factors have weak explanatory ability in China.). In conclusion, the sub-sample results in Table 7 show that, markets co-move strongly during the periods of market downturns (Das et al. 2018), and MFB's forecasting ability fades.

## A risk perspective?

The MFB that we define from an investor psychology perspective differs from the difference in downside and upside beta (Delta Beta) defined by Ang et al. [12] only by the inclusion of the absolute value. Therefore, can the delta beta defined from a risk perspective also predict future stock returns? In other words, is the risk factor the cause of MFB predicting future stock returns? One simple way to answer this question is to examine whether the difference in downside / upside betas constructed from a risk perspective can predict future stock returns. If

**Table 8. Additional statistical description.**

|  | MFB | Delta Beta | MFB Zero | MFB Mu | Mean | STD |
|---|---|---|---|---|---|---|
| MFB | 1.00 | 0.36 | >0.99 | 0.96 | 1.12 | 0.49 |
| Delta Beta | 0.36 | 1.00 | 0.36 | 0.35 | 0.23 | 0.39 |
| MFB Zero | >0.99 | 0.36 | 1.00 | 0.96 | 1.12 | 0.49 |
| MFB Mu | 0.96 | 0.35 | 0.96 | 1.00 | 1.10 | 0.47 |

the difference in downside / upside betas constructed from a risk perspective cannot predict future stock returns, while MFB constructed from a behavioral finance perspective including the absolute value can, it suggests that MFB is indeed not a proxy variable for risk.

Firstly, in Table 8, we present a simple descriptive analysis of the delta beta statistic. The mean value is 0.23 and the variance is 0.39, with a correlation coefficient of 0.36 to MFB. Results show that most of the monthly downside betas are larger than their upside counterparts. However, the correlation coefficient is less than 0.5, suggesting that there is still a pronounced deviation between MFB and delta beta.

Furthermore, we conducted firm-level Fama-MacBeth regressions [15] to explore the predictive power of Delta Beta for one-month ahead stock returns. Results are reported in Table 9. The OLS method yielded a coefficient of -0.42 with a t-statistic of -0.26, while the WLS method produced a coefficient of -0.14 with a t-statistic of -1.21. These findings indicate that Delta Beta, constructed from a risk perspective, does not possess predictive power for future stock returns. This result is consistent with the conclusion reported in Levi and Welch [13]. However, as we demonstrated in the previous section, MFB constructed from an investor sentiment bias can predict future stock returns effectively. These results indicate that MFB captures the prediction of future stock returns using investor sentiment bias, rather than risk.

## Reference point

According to Junior et al. [8], framework bias is related to prospect theory, which suggests that investment decisions are influenced by reference points (thresholds). Liu [40] gives a summary regarding the thresholds for the downside beta used in the literature. Therefore, in this subsection, we analyze the issue of reference point selection in the construction of the MFB process. In fact, Ang et al. [12] mainly focus on three reference points: 1) riskless rate; 2) zero rate of return; and 3) average market returns (Mu). However, we only considered the riskless rate in the previous sections. So, in this subsection, we construct new versions of the MFB, named MFB Zero and MFB Mu, using zero rate of return and average market returns on each month as reference points, respectively. Through this, we aim to further discuss the issue of reference point selection.

In fact, due to the small size of the riskless rate on a daily basis, MFB and MFB Zero should be highly correlated. As confirmed by the results in Table 8 from the previous section, the correlation coefficient between MFB and MFB Zero is greater than 0.99, indicating that the choice of reference point between riskless rate and zero rate of return is almost identical. Additionally, the results in Table 8 also show a high correlation coefficient of 0.96 between MFB and MFB Mu, indicating that using average market returns as the reference point has a strong correlation with using riskless rate as the reference point. Our results are almost identical to Ang et al.'s conclusions: These betas calculated by different reference points exhibit a correlation greater than 0.96.

Finally, we conducted firm-level Fama-MacBeth regressions based on ordinary least squares (OLS) and weighted least squares (WLS, Asparouhova et al. [41]), respectively, to

**Table 9. Additional statistical description.**

| | OLS | | | WLS | | |
|---|---|---|---|---|---|---|
| Delta Beta | -0.42 | | | -0.14 | | |
| | (-0.26) | | | (-1.21) | | |
| MFB Zero | | -2.22‡ | | | -0.22‡ | |
| | | (-5.56) | | | (-6.74) | |
| MFB Mu | | | -2.30‡ | | | -0.24‡ |
| | | | (-5.86) | | | (-6.45) |
| Beta | -3.84‡ | -3.84‡ | -3.84‡ | 0.36‡ | 0.37‡ | 0.36‡ |
| | (-3.78) | (-3.78) | (-3.78) | (3.89) | (4.27) | (4.28) |
| Size | 1.54† | 1.41† | 1.39† | -0.45‡ | -0.45‡ | -0.45‡ |
| | (2.49) | (2.28) | (2.25) | (-3.65) | (-3.61) | (-3.62) |
| BM | -0.7 | -0.67 | -0.66 | 0.1 | 0.09 | 0.09 |
| | (-1.48) | (-1.43) | (-1.41) | (1.39) | (1.22) | (1.18) |
| Str | 1.83‡ | 1.90‡ | 1.90‡ | -0.07 | -0.06 | -0.06 |
| | (2.69) | (2.77) | (2.78) | (-1.35) | (-1.3) | (-1.27) |
| MoM | 4.50 | 4.83 | 4.80 | 0.16† | 0.17† | 0.17† |
| | (1.59) | (1.60) | (1.60) | (2.47) | (2.57) | (2.55) |
| Illiq | 0.52 | 0.44 | 0.37 | 0.24‡ | 0.24‡ | 0.24‡ |
| | (0.38) | (0.51) | (0.42) | (3.1) | (3.17) | (3.16) |
| Coskew | 0.79 | 0.17 | 0.18 | -0.03 | 0.07 | 0.06 |
| | (0.75) | (0.22) | (0.23) | (-0.19) | (0.68) | (0.62) |
| Betadown | -3.61‡ | -3.37‡ | -3.35‡ | 0.11 | 0.04 | 0.04 |
| | (-8.81) | (-8.04) | (-8.04) | (0.93) | (0.37) | (0.47) |
| Voldu | -3.12‡ | -2.86‡ | -2.86‡ | -0.35‡ | -0.33‡ | -0.32‡ |
| | (-8.00) | (-7.23) | (-7.18) | (-7.12) | (-6.51) | (-6.51) |
| VaR | 3.12‡ | 3.24‡ | 3.19‡ | -0.34‡ | -0.31‡ | -0.31‡ |
| | (4.01) | (4.78) | (4.78) | (-6.38) | (-5.89) | (-5.82) |
| Adj. R2 | 0.09 | 0.09 | 0.09 | 0.09 | 0.09 | 0.09 |

Note: This table reports the results of the Fama-MacBeth cross-sectional regressions of individual firms for future 1 month on the control variables measured in month $t$. The columns on the left show the results of ordinary least squares. The columns on the right present results estimated for one-month-ahead returns using the weighted least squares (WLS) methodology of Asparouhova et al. [41], where each observed return is weighted by one plus the observed prior return on the stock. Newey and West [33] t-statistics are given in parentheses.

‡: Significant at 1%.

†: Significant at 5%.

*: Significant at 10%.

explore the predictive power of MFB Zero and MFB Mu for one-month ahead stock returns. Results are reported in Table 9.

The results in Table 9 show that regardless of whether the zero rate of return or average market returns are used as the reference point, and whether OLS or WLS is used, MFB still has a significant predictive power for stock returns in the next month. For instance, when using the zero rate of return as the reference point, the Fama-MacBeth regression coefficient of MFB Zero based on OLS is -2.22, with a t-statistic of -5.56, while that based on WLS is -0.22, with a t-statistic of -6.74. Moreover, we also found an interesting phenomenon: the sign of the Fama-MacBeth regression coefficients of Beta and Betadown based on OLS or WLS differs, while our MFB has a consistent sign. This further suggests that our MFB is more stable than Beta and

Betadown in predicting stock returns when the prior return on the stock is taken into account. In summary, the predictive power of MFB based on different reference points is robust.

## Conclusion

In this paper, we propose that investors may exhibit a framing effect in the return-risk tradeoff under different frameworks of aggregate market losses and profits, which is defined as the market framing bias (MFB). We measure the MFB of an individual stock using the absolute difference between the betas in the up and down markets, and we also explore the predictive power of the MFB on future stock returns in the cross-section. More specifically, we find the following:

First and foremost, the paper finds that MFB is able to predict future stock returns in the cross-section. By going long a portfolio of the stocks with the lowest MFB and short selling a portfolio of the stocks with the highest MFB, we find that there is a significant negative $\tau$-month-ahead excess return, where $\tau$ ranges from 1 month to 12 months. In other words, the MFB is able to predict the excess return from 1 month to 12 months. Moreover, we show that, after controlling for various firm-specific characteristics, the predictive power of MFB diminishes as future returns are predicted from 1 month to 12 months. Last but not least, various asset pricing factors (FF5, SL3, UMD, and SENT) do not eliminate the MFB's predictive power. We confirm our conclusions by replacing the seminal FF5 factor with the popular Q4 asset pricing factor and performing a bivariate portfolio analysis, as well as by using out-of-sample tests.

Our findings exhibit robustness across the following dimensions: Subsample analysis conducted on boom and recession periods, comparison of Delta Beta's predictive ability from a risk perspective, and varying reference points utilized in MFB construction. Ongoing changes in the stock market structure also present new challenges for researchers. For example, we may construct an asset pricing factor based on the MFB and study if this factor exhibits explanatory power on the cross-section of stock returns.

## Supporting information

**S1 Data.**
(ZIP)

## Author Contributions

**Conceptualization:** Jun Xie.

**Data curation:** Baohua Zhang, Bin Gao.

**Formal analysis:** Jun Xie, Bin Gao.

**Funding acquisition:** Jun Xie, Bin Gao.

**Methodology:** Jun Xie, Bin Gao.

**Software:** Jun Xie, Bin Gao.

**Supervision:** Jun Xie.

**Validation:** Baohua Zhang.

**Writing – original draft:** Baohua Zhang, Bin Gao.

**Writing – review & editing:** Jun Xie.

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
