## [Editor Report · Decision Letter 0]

29 Mar 2023

PONE-D-23-05212Market framing bias and cross-sectional stock returnsPLOS ONE

Dear Dr. Gao,

Thank you for submitting your manuscript to PLOS ONE. After careful consideration, we feel that it has merit but does not fully meet PLOS ONE’s publication criteria as it currently stands. Therefore, we invite you to submit a revised version of the manuscript that addresses the points raised during the review process.

We look forward to receiving your revised manuscript.

Kind regards,

Bartosz Gebka, PhD

Academic Editor

PLOS ONE

Journal Requirements:

4. Please ensure that you include a title page within your main document. You should list all authors and all affiliations as per our author instructions and clearly indicate the corresponding author.

Additional Editor Comments:

I thank the authors for submitting their work to PLOS ONE.

Before I would send the paper to reviewers, to give the paper a fair chance, I would like the authors to address the following issues:

1. theoretical rationale. It is not clear to me how exactly we can use the up vs down beta as a measure of the framing bias among investors. This needs to be explained in more detail. the authors say "Glascock and Lu-Andrews (2018) discover that the beta coefficients of the

market are different under different market states, which is probably due to the framing effect.

This evidence reveals that the framing effect in a stock market can be represented by the

difference between beta in different market statuses (up or down markets), which inspires us to

use the absolute difference between betas in the up and down markets to measure the framing

effect." The key, and worrying, term is PROBABLY here. As a rationale it is too weak. the reader needs to be reasonably strongly convinced that the measure used is most likely measuring what it is suppose to measure.

2. one would expect an empirical investigation into if the proposed factor is a priced risk factor in equilibrium, this does not seem to have been done? Ie accounting for the well known factors (eg FF5) is the proposed effect still attracting a significant risk premium?

3. there is a substantial literature on betas in up vs down markets, one would expect that some(alternative) explanations have been proposed for the betas difference. how does this paper differentiates between different explanations for differences in up vs down betas?

---

## [Author Response · Author response to Decision Letter 0]

23 Apr 2023

Responses to the reviewer

PLOS ONE

We respond to the comments, shown in red, below. We use the letter R to indicate our response.

Journal Requirements:

R: We re-formatted the paper according to the PLOS ONE style template to meet the requirements of PLOS ONE.

R: We asked a professor (HZ. Zeng) who had published papers in Management Science to help us with copyedit. And we hired Editsprings to make numerous revisions to the paper, including language usage, spelling, and grammar. Details can be queried from the supporting information file.

We have also uploaded the supporting information file and the new manuscript file as requested.

R: Upon re-submitting our revised manuscript, we upload our study’s minimal underlying data set as Supporting Information files, please check it.

4. Please ensure that you include a title page within your main document. You should list all authors and all affiliations as per our author instructions and clearly indicate the corresponding author.

R: Upon re-submitting our revised manuscript, we upload a title page within the document, which lists all authors and all affiliations as per our author instructions and we clearly indicate the corresponding author.

Additional Editor Comments:

I thank the authors for submitting their work to PLOS ONE.

Before I would send the paper to reviewers, to give the paper a fair chance, I would like the authors to address the following issues:

1. theoretical rationale. It is not clear to me how exactly we can use the up vs down beta as a measure of the framing bias among investors. This needs to be explained in more detail. the authors say "Glascock and Lu-Andrews (2018) discover that the beta coefficients of the market are different under different market states, which is probably due to the framing effect.

This evidence reveals that the framing effect in a stock market can be represented by the

difference between beta in different market statuses (up or down markets), which inspires us to use the absolute difference between betas in the up and down markets to measure the framing effect." The key, and worrying, term is PROBABLY here. As a rationale it is too weak. the reader needs to be reasonably strongly convinced that the measure used is most likely measuring what it is suppose to measure.

R: We give three reasons in Section 2 (Quantification of MFB) to describe why we can use the absolute difference between betas in the up and down markets to measure the market framing effect. And in the revised version, we improved these three reasons to make them better express our views. The revised version also improved the description of “PROBABLY”.

We put some of the improved excerpts below: 

First, the up or down market provides a “natural frame” to characterize the investment climate. Kahneman (2003) points out that the framing effect is the anomaly that extensionally equivalent descriptions lead to different choices by altering the relative salience of different aspects of the problem (different environments). For investors, as observed by Glaser et al. (2007), a positive frame (profit-making environment) or a negative frame (loss-making environment) may naturally indicate the up or down market. In other words, our definition of such a bias as MFB is similar to Kahneman's (2003) definition of framing bias, except that our bias is produced under a different market framework. That's why we call it MFB, instead of just calling it framing bias.

Second, we use the difference between the betas in the up and down markets to measure the behavioral bias of investors. Glascock and Lu-Andrews (2018) found that the market beta coefficients changed under different market states, which could be attributed to the framing effect. This evidence reveals that the framing effect in a stock market can be reflected by the difference between beta in different market situations (up or down markets), which inspires us to use the absolute difference between betas in the up and down markets to measure the framing effect.

Finally, we take the absolute value, because the framing effect, as defined by Kahneman (2003), has no positive or negative direction. It occurs when different descriptions of the framework lead to irrational decisions of investors. According to CAPM theory (Sharpe, 1964; Lintner, 1965), rational investors should have the same beta in both up and down markets. If the beta values differ across market status (frameworks), it reflects the existence of a framing effect for investors.”

2. one would expect an empirical investigation into if the proposed factor is a priced risk factor in equilibrium, this does not seem to have been done? Ie accounting for the well known factors (eg FF5) is the proposed effect still attracting a significant risk premium?

R: To the best of our knowledge, previous studies on framing effect focused on discussing the existence of framing effect. We are the first to measure the “market framing bias” for individual stocks, that seeks to use MFB as an individual stock characteristic and explore whether it can generate anomalies. Building a pricing factor based on MFB will be the topic of our next paper, because it is only possible to further study whether it is a common pricing factor after discussing the anomaly of MFB clearly. If we put the two topics in one paper, the paper would be too long, and the theme would not be clear enough. Therefore, in this paper, we mainly discuss whether “FMB” as an individual stock characteristic will generate excess returns that cannot be explained by pricing factors such as FF5, Q4, SL2 factors. The excess return reported in Table 3 and Table 4 of the paper is the average of the excess returns of the corresponding portfolios, that is, the risk premium of the corresponding portfolios (see Bodie et al. 2022, Essentials of Investment, McGraw Hill LLC). The alpha then calculated is the risk premium adjusted for factors such as FF5. The source of the method in this paper can refer to Empirical Asset Pricing (Bali, et al. 2016, Published by John Wiley & Sons, Inc., Hoboken, New Jersey).

3. there is a substantial literature on betas in up vs down markets, one would expect that some(alternative) explanations have been proposed for the betas difference. how does this paper differentiates between different explanations for differences in up vs down betas?

R: There is a large amount of literature that studies the beta coefficients in rising and falling markets, but they mainly explain the difference in beta coefficients by the different risk preferences of investors in rising and falling markets. This paper is different from them, and defines the difference in investors’ return predictions (beta) under different scenarios (frames) in rising and falling markets as MFB. The framing effect is an anomaly that extensionally equivalent descriptions lead to different choices by altering the relative salience of different aspects of the problem (Kahneman, 2003). FB contains many aspects, and this paper only analyzes it from the different frames of market rising and falling, so it is defined as MFB. Before this, the frame effect was mainly measured by questionnaire surveys, and this paper gives an objective method to measure it using public trading data, which provides important reference significance for the quantitative research of behavioral finance, and has great innovation.

---

## [Decision Letter · Decision Letter 1]

10 Jul 2023

PONE-D-23-05212R1Market framing bias and cross-sectional stock returnsPLOS ONE

Dear Dr. Gao,

Thank you for submitting your manuscript to PLOS ONE. After careful consideration, we feel that it has merit but does not fully meet PLOS ONE’s publication criteria as it currently stands. Therefore, we invite you to submit a revised version of the manuscript that addresses the points raised during the review process.

We look forward to receiving your revised manuscript.

Kind regards,

Bartosz Gebka, PhD

Academic Editor

PLOS ONE

Additional Editor Comments:

Thank you for submitting your manuscript to PLOS ONE. We have now heard from two reviewers who are experts in this filed and, while they see many merits in the manuscript, they also ask for some clarifications and revisions. Hence we would like to offer you a revise-and-resubmit. Should you agree to take up this offer, please revise the paper along the lines raised by the reviewers, and also provide a detailed document on how their points have been addressed, upon resubmission.

I hope these comments will be helpful and will help to improve the quality of the manuscript.

Best regards

Reviewers' comments:

Reviewer's Responses to Questions

**Comments to the Author**

1. If the authors have adequately addressed your comments raised in a previous round of review and you feel that this manuscript is now acceptable for publication, you may indicate that here to bypass the “Comments to the Author” section, enter your conflict of interest statement in the “Confidential to Editor” section, and submit your "Accept" recommendation.

Reviewer #1: (No Response)

Reviewer #2: All comments have been addressed

2. Is the manuscript technically sound, and do the data support the conclusions?

Reviewer #1: Partly

Reviewer #2: Partly

3. Has the statistical analysis been performed appropriately and rigorously? 

Reviewer #1: Yes

Reviewer #2: (No Response)

4. Have the authors made all data underlying the findings in their manuscript fully available?

Reviewer #1: Yes

Reviewer #2: Yes

5. Is the manuscript presented in an intelligible fashion and written in standard English?

Reviewer #1: Yes

Reviewer #2: (No Response)

6. Review Comments to the Author

Reviewer #1: This paper constructs a market framing bias (MFB) measure to predict the cross-sectional stock returns. The idea is intuitive, and the paper is easy to follow. However, I have the following doubts.

Most importantly, contributions in this paper do not sound enough. The authors define the MFB measure as the absolute value of the difference between beta in times of market returns above riskless rate and beta in times of market returns falling below riskless rate. Computing beta in times of market returns falling below riskless rate has been proposed by Hogan & Warren as early as 1974. Taking the difference between downside beta and upside beta is also not new. Ang et al. (2006) has done this and explored its predictive power.

Although the downside / upside betas used in Ang et al. (2006)’s main analyses are defined relative to average market returns, they explored the correlation between these betas and betas defined relative to riskless rate and zero rate of return in the robustness test. These betas exhibit a correlation greater than 0.96. In this paper, the authors examined the correlation between their MFB measure and Ang’s downside beta but the correlation is only 0.39. First, it is more appropriate to compare MFB with difference between Ang’s downside and upside betas. Second, it is better to explain what causes the discrepancy between the two papers.

The authors also need to explain why they compute downside betas relative to riskless rate. Economists have used various thresholds for computing downside betas. See Liu (2023) for a summary regarding the thresholds used in the literature. Given Ang has already explored the predictive power of the difference between downside and upside betas, it becomes especially crucial for the authors to explain why betas computed using riskless rate works better than Ang’s use of average market returns.

The authors highlight that they take the absolute value of the difference between downside and upside betas as MFB. As we know, stocks tend to co-move more strongly when the market declines. Thus, downside betas tend to be greater than upside betas. I am curious about the necessity to take the absolute value.

In addition, upside betas have been found insignificant. See Ang et al (2006) for example. It is important for the authors to explain why MFB is a better predictor than the downside beta computed relative to riskless rate alone.

References

Ang, A., Chen, J., & Xing, Y. (2006). Downside risk. The review of financial studies, 19(4), 1191-1239.

Hogan, W. W., & Warren, J. M. (1974). Toward the development of an equilibrium capital-market model based on semivariance. Journal of Financial and Quantitative Analysis, 9(1), 1-11.

Liu, J. (2023). A novel downside beta and expected stock returns. International Review of Financial Analysis, 85, 102455.

Reviewer #2: Reviewer comments

Rationale for MFB as a Measure of Framing Effect: The manuscript asserts that MFB is a measure of the framing effect observed in the stock market. However, the justification for this claim is not thoroughly discussed. The authors briefly mention that the framing effect in behavioural finance is a cognitive bias based on whether investors are in a profit-making or loss-making environment. It would be beneficial to provide a more comprehensive explanation of how MFB captures this framing effect and the underlying mechanisms through which it influences investor decision-making

Data

The sample period of January 2000 to December 2019 appears to be extensive, allowing for a comprehensive analysis of the research question. The inclusion criterion of stocks traded for at least 36 months during the sample period is reasonable and helps ensure a sufficient number of observations. With 3804 stocks and nearly 500,000 firm-month observations, the sample size is substantial and provides a solid foundation for statistical analysis. Controlling for firm-specific characteristics is crucial in studying cross-sectional returns, and the manuscript lists several variables used for this purpose. The inclusion of beta, size, book-to-market equity ratio, short-term reversal, momentum return, illiquidity, co-skewness, downside beta, VOLDU, and VaR provides a comprehensive set of factors that may affect expected stock returns.

1. Definition of MFB: The manuscript introduces MFB as the bias of the risk-return trade-off between up and down markets. While the rationale for this definition is provided, it would be helpful to have a more explicit connection between MFB and the framing effect. The authors briefly mention that MFB reflects how investor expectations differ in the gain/loss framework, but a deeper discussion on how MFB captures framing effects and cognitive biases would enhance the understanding of the concept.

2. Model Specification: Equation (1) represents the model used to estimate the difference between betas in up and down markets. However, the authors should provide a more detailed explanation of the variables and parameters in the equation. Specifically, they should clarify the definitions of R, f, M, and ε, and explain how these variables are calculated or sourced. Providing more clarity on the model specification would improve the reproducibility and transparency of the methodology.

3. Justification for Absolute Difference: The authors justify the use of the absolute difference between betas by stating that the framing effect has no positive or negative direction. While this reasoning is understandable, it would be valuable to discuss potential drawbacks or limitations of this approach. Are there any scenarios or implications where considering the direction of the difference could provide additional insights? Discussing alternative approaches and their respective advantages and disadvantages would strengthen the methodology.

4. Empirical Evidence and Validation: The manuscript states that MFB may convey information useful for predicting future stock returns. However, no empirical evidence or validation of this claim is provided. It is important to include results from statistical tests or regression analyses that demonstrate the predictive power of MFB in forecasting future returns. Additionally, discussing the robustness of the findings and potential confounding factors would enhance the credibility of the methodology.

5. Comparison with Previous Studies: The authors briefly mention Levi and Welch (2020) as a previous study, but more discussion and comparison with relevant literature is needed. How does the proposed methodology differ from and build upon existing approaches to measuring framing effects and behavioural biases in financial markets? Highlighting the unique contributions and advantages of the current methodology would strengthen the manuscript's originality.

In summary, while the manuscript presents a methodology for quantifying MFB, there are several areas that require further clarification, empirical evidence, and comparative analysis. Addressing these points would improve the overall strength and reliability of the methodology section.

1. Methodological Details: The manuscript lacks detailed explanations of the specific calculations and data sources used to measure MFB and construct the decile portfolios. Providing step-by-step explanations of the calculations, including the formulas used and the specific databases or sources of data, would enhance the replicability of the study.

2. Interpretation of Results: While the summary table provides the excess returns and abnormal returns for each decile portfolio, the manuscript lacks a comprehensive interpretation and discussion of the findings. It is essential to provide a deeper analysis of the results, including their statistical significance, economic significance, and potential implications for investors. Additionally, connecting the findings back to the literature on the framing effect and behavioral finance would strengthen the manuscript's contribution.

3. Bivariate Analysis: The manuscript briefly mentions a bivariate portfolio analysis but does not provide detailed results or discussion of this analysis. It would be valuable to present the findings of the bivariate analysis and explore the relationship between MFB and other firm-specific characteristics or behavior-related asset pricing factors. This would provide a more comprehensive understanding of the factors that influence MFB and their impact on stock returns.

4. Limitations and Robustness: The manuscript does not address potential limitations of the methodology or conduct robustness tests to assess the stability and reliability of the findings. It is crucial to acknowledge any limitations of the study, such as data limitations or assumptions made, and discuss how these limitations may affect the interpretation of the results. Additionally, conducting sensitivity analyses or alternative model specifications would enhance the robustness of the findings.

7. PLOS authors have the option to publish the peer review history of their article (what does this mean?). If published, this will include your full peer review and any attached files.

Reviewer #1: No

Reviewer #2: No

---

## [Author Response · Author response to Decision Letter 1]

27 Jul 2023

Responses to the reviewers

PLOS ONE

We have extended the length of our paper in order to provide better responses to the comments from the reviewers. We respond to the comments, shown in red, below. We use the letter R to indicate our response.

Reviewers' comments:

Reviewer #1: This paper constructs a market framing bias (MFB) measure to predict the cross-sectional stock returns. The idea is intuitive, and the paper is easy to follow. However, I have the following doubts.

Most importantly, contributions in this paper do not sound enough. The authors define the MFB measure as the absolute value of the difference between beta in times of market returns above riskless rate and beta in times of market returns falling below riskless rate. Computing beta in times of market returns falling below riskless rate has been proposed by Hogan & Warren as early as 1974. Taking the difference between downside beta and upside beta is also not new. Ang et al. (2006) has done this and explored its predictive power.

R: On the one hand, although Ang et al. (2006) calculated the difference (Delta Beta) between downside beta and upside beta and explored its predictive power, subsequent studies, such as Levi and Welch (2020), did not confirm its predictive power. However, our constructed MFB, despite being only a minor improvement over Delta Beta, significantly and consistently predicts future stock returns, which Delta Beta cannot.

On the other hand, Ang et al. (2006) constructed the difference of downside/upside beta (Delta Beta) from the perspective of risk, considering the reference points of downward/upward market defined relative to the average market returns, riskless rate, and zero rate of return. Levi and Welch (2020) mentioned the difference between downside beta and upside beta relative to the average market returns. However, it is important to note that the papers by Ang et al. (2006) and Levi and Welch (2020) interpreted beta as risk and explained their impact on future returns from the perspective of risk (market risk exposures are time-vary). This paper, on the other hand, is the first to explore the significance of the difference between upside and downside beta from the perspective of investor psychology and behavioral biases, which is the biggest contribution of this paper. It is precisely because we calculate upside and downside beta from the perspective of investor behavior biases and construct MFB that our approach differs from that of Ang et al. (2006) and Levi and Welch (2020): our calculation of MFB using one month of data from the perspective of short-term investor behavior biases can significantly negatively affect future stock returns. 

On Page 3 of the revised version, we emphasize the contribution of this paper, and on Page 6 (Line 149-160), we emphasize the significance of investor psychological bias in taking the absolute difference of the upside beta and downside beta, and discuss the "short-term" reasons for taking monthly daily data for calculation: Investors' psychological biases are often short-lived. This is significantly different from Ang et al. (2006) taking the data of the past year to calculate downside beta and upside beta. In fact, Levi and Welch's (2020) research shows that taking one year's data to calculate the Absolute delta beta difference between downside beta and upside beta does not predict future stock returns. However, through the analysis of this paper, we take one month's data from the perspective of investors' short-term behavior deviation to calculate that MFB can significantly negatively affect the future returns of stocks.

Although the downside / upside betas used in Ang et al. (2006)’s main analyses are defined relative to average market returns, they explored the correlation between these betas and betas defined relative to riskless rate and zero rate of return in the robustness test. These betas exhibit a correlation greater than 0.96. In this paper, the authors examined the correlation between their MFB measure and Ang’s downside beta but the correlation is only 0.39. First, it is more appropriate to compare MFB with difference between Ang’s downside and upside betas. Second, it is better to explain what causes the discrepancy between the two papers.

R: We are grateful for the reviewer's suggestion to compare our MFB with Ang et al.'s downside and upside betas. In the revised version of our paper, we conducted robustness tests using different reference points (relative to average market returns and to zero rate of return) for calculating MFB, and demonstrated the robustness of our study (see details on Page 21, Table 8). We found that the correlation coefficients between MFB and MFB Zero (defined relative to zero rate of return), as well as between MFB and MFB Mu (defined relative to average market returns), are also greater than 0.96.

Ang et al. (2006) indicates that the downside and upside betas, which are defined relative to average market returns, riskless rate, and zero rate of return, exhibit a correlation greater than 0.96. For instance, the correlation between the downside beta defined relative to average market returns with the downside beta defined relative to riskless rate is 0.971. However, as our paper's MFB is defined as the absolute difference between the downside and upside betas, the correlation between our MFB and downside beta of Ang et al. (2006) has decreased to 0.39, which is the performance in accordance with the absolute value. The main reason for the difference between the two papers is that we calculate the absolute difference from the perspective of investors' short-term behavior deviation.

The authors also need to explain why they compute downside betas relative to riskless rate. Economists have used various thresholds for computing downside betas. See Liu (2023) for a summary regarding the thresholds used in the literature. Given Ang has already explored the predictive power of the difference between downside and upside betas, it becomes especially crucial for the authors to explain why betas computed using riskless rate works better than Ang’s use of average market returns.

R: Thank you for the reviewer's feedback. We did overlook the issue of selecting thresholds for distinguishing between up and down markets in the original manuscript. Considering that Ang et al. (2006) mainly focused on three reference points: 1) riskless rate; 2) zero rate of return; and 3) average market returns (Mu), we supplement the discussion of the last two reference points (thresholds) in the revised version's robustness test (Page 21-23). The results show that our conclusion is robust, and the choice of references does not affect the predictive ability of MFB. Furthermore, we explain on Page 11 of the revised version why our MFB performs better than Ang et al.'s Delta Beta in terms of prediction: Firstly, we take the absolute value of Delta Beta from the perspective of investors' short-term behavioral biases, while it is hard to explain why Ang et al. (2006) take absolute values from the perspective of risk. Secondly, when calculating the upside/downside betas, we used daily data within each month based on investors' short-term behavioral biases, whereas Ang et al. (2006) used daily data from the past year.

The authors highlight that they take the absolute value of the difference between downside and upside betas as MFB. As we know, stocks tend to co-move more strongly when the market declines. Thus, downside betas tend to be greater than upside betas. I am curious about the necessity to take the absolute value.

R: Indeed, from a risk perspective, stocks tend to co-vary more strongly when the market falls, with down beta being usually greater than up beta. However, it's necessary to take the absolute value for two main reasons: firstly, theoretically, the definition of framing bias does not involve direction, so we define MFB as the absolute value (ignoring the direction issue); secondly, our stability test on the revised draft (Table 8) also shows that the correlation coefficient between taking the absolute value (MFB) and not taking the absolute value (Delta Beta) is 0.36, which is not very high. Furthermore, most importantly, in Table 9, we found that not taking the absolute value (Delta Beta) does not have the ability to predict the future returns of stocks. Therefore, it's necessary to take the absolute value.

In addition, upside betas have been found insignificant. See Ang et al (2006) for example. It is important for the authors to explain why MFB is a better predictor than the downside beta computed relative to riskless rate alone.

R: In fact, not only has upside beta been found to be irrelevant, but Levi and Welch (2020) also found that stocks with higher down-betas ex ante do not earn higher average rates of return ex post. This is one of the reasons that motivated the research in this article: whether there exist asymmetrical beta-related anomalies. However, in constructing MFB, this article primarily considers investors' short-term framing bias. Therefore, the main perspective when writing the paper is from the angle of investors' short-term behavioral bias. As for why MFB is a better predictor than the down beta calculated independently with the risk-free rate, the absolute delta beta exactly captures investors' framing effect bias, which can often lead to investment losses. Ang et al. (2006) and Levi and Welch (2020) started from the perspective of risk and were unable to capture such a negative correlation relation. Therefore, based on this analysis, we believe that MFB is a better predictor than the down beta calculated independently with the risk-free rate.

References

Ang, A., Chen, J., & Xing, Y. (2006). Downside risk. The review of financial studies, 19(4), 1191-1239.

Hogan, W. W., & Warren, J. M. (1974). Toward the development of an equilibrium capital-market model based on semivariance. Journal of Financial and Quantitative Analysis, 9(1), 1-11.

Liu, J. (2023). A novel downside beta and expected stock returns. International Review of Financial Analysis, 85, 102455.

 

Reviewer #2: Reviewer comments

Rationale for MFB as a Measure of Framing Effect: The manuscript asserts that MFB is a measure of the framing effect observed in the stock market. However, the justification for this claim is not thoroughly discussed. The authors briefly mention that the framing effect in behavioural finance is a cognitive bias based on whether investors are in a profit-making or loss-making environment. It would be beneficial to provide a more comprehensive explanation of how MFB captures this framing effect and the underlying mechanisms through which it influences investor decision-making

Data

The sample period of January 2000 to December 2019 appears to be extensive, allowing for a comprehensive analysis of the research question. The inclusion criterion of stocks traded for at least 36 months during the sample period is reasonable and helps ensure a sufficient number of observations. With 3804 stocks and nearly 500,000 firm-month observations, the sample size is substantial and provides a solid foundation for statistical analysis. Controlling for firm-specific characteristics is crucial in studying cross-sectional returns, and the manuscript lists several variables used for this purpose. The inclusion of beta, size, book-to-market equity ratio, short-term reversal, momentum return, illiquidity, co-skewness, downside beta, VOLDU, and VaR provides a comprehensive set of factors that may affect expected stock returns.

1. Definition of MFB: The manuscript introduces MFB as the bias of the risk-return trade-off between up and down markets. While the rationale for this definition is provided, it would be helpful to have a more explicit connection between MFB and the framing effect. The authors briefly mention that MFB reflects how investor expectations differ in the gain/loss framework, but a deeper discussion on how MFB captures framing effects and cognitive biases would enhance the understanding of the concept.

R: In the revised draft on Page 4-5, we described how MFB captures framing effects and cognitive biases:

We attempt to measure the MFB in terms of the absolute difference between the betas in the up and down markets, which is attributed to the following reasons:

First, the up or down market provides a “natural frame” to characterize the investment climate. Kahneman (2003) points out that the framing effect is the anomaly that extensionally equivalent descriptions lead to different choices by altering the relative salience of different aspects of the problem (different environments). For investors, as observed by Glaser et al. (2007), a positive frame (profit-making environment) or a negative frame (loss-making environment) may naturally indicate the up or down market. In other words, our definition of such a bias as MFB is similar to Kahneman's (2003) definition of framing bias, except that our bias is produced under a different market framework. That's why we call it MFB, instead of just calling it framing bias.

Second, we use the difference between the betas in the up and down markets to measure the behavioral bias of investors. Glascock and Lu-Andrews (2018) found that the market beta coefficients changed under different market states, which could be attributed to the framing effect. This evidence reveals that the framing effect in a stock market can be reflected by the difference between beta in different market situations (up or down markets), which inspires us to use the difference between betas in the up and down markets to measure the framing effect. 

Finally, we take the absolute value, because the framing effect, as defined by Kahneman (2003), has no positive or negative direction. It occurs when different descriptions of the framework lead to irrational decisions of investors. According to CAPM theory (Sharpe, 1964; Lintner, 1965), rational investors should have the same beta in both up and down markets. If the beta values differ across market status (frameworks), it reflects the existence of a framing effect for investors.

2. Model Specification: Equation (1) represents the model used to estimate the difference between betas in up and down markets. However, the authors should provide a more detailed explanation of the variables and parameters in the equation. Specifically, they should clarify the definitions of R, f, M, and ε, and explain how these variables are calculated or sourced. Providing more clarity on the model specification would improve the reproducibility and transparency of the methodology.

R: The revised draft provides detailed explanations for Model (1)

Where R_(i,t) is the return of stock i at day t in month K, R_(f,t) is the risk-free interest rate at day t, R_(M,t) is the return of the whole stock market on day t, R_(M,t) 〖-R〗_(f,t) is the market premium (market factor of CAPM), α_i is the intercept, β and β ~ the regression coefficient，ε_t is the residual. 

To improve the reproducibility and transparency of the methodology, in the revised draft on Page 6-7, we also provided a detailed explanation for the data source:

We collect the sample data for all A-shares (traded in the Shanghai Stock Exchange and Shenzhen Stock Exchange, excluding SSE STAR Market). Daily and monthly stock market data used in this paper are from the RESSET database, except for the following data that come from China Stock Market & Accounting Research Database (CSMAR): momentum (UMD; Carhart, 1997), sentiment (SENT; Baker and Wurgler, 2006), monthly excess returns on the market (MKT), size (SMB), value (HML), investment (CMA) and profitability (RMW) factors of Fama and French (2015). The sample period is from January 2000 to December 2019, and stocks must have been traded for at least 36 months during the sample period. The final sample contains 3804 stocks and a total of nearly 500,000 firm-month observations.

3. Justification for Absolute Difference: The authors justify the use of the absolute difference between betas by stating that the framing effect has no positive or negative direction. While this reasoning is understandable, it would be valuable to discuss potential drawbacks or limitations of this approach. Are there any scenarios or implications where considering the direction of the difference could provide additional insights? Discussing alternative approaches and their respective advantages and disadvantages would strengthen the methodology.

R: The non-absolute value method primarily discusses the predictive power of downside beta from a risk perspective, which has already been analyzed in Ang et al. (2006) and Levi and Welch (2020). However, Levi and Welch (2020) analyzed the non-absolute value method from the perspective of risk and found that it cannot significantly predict future stock returns. We also discussed in Table 9 of the revised draft that the non-absolute value method (Delta Beta) does not have predictive ability.

Of course, our method of taking absolute values also has flaws. For example, it ignores the fact that stocks tend to exhibit stronger co-movement during market downturns (Das et al. 2018), and the issue of how to determine the threshold (reference point) for determining whether the market is in an upturn or downturn. Therefore, in Section 5 (Robustness tests) of the revised draft, we strengthened the stability discussion and analyzed the above-mentioned issues.

4. Empirical Evidence and Validation: The manuscript states that MFB may convey information useful for predicting future stock returns. However, no empirical evidence or validation of this claim is provided. It is important to include results from statistical tests or regression analyses that demonstrate the predictive power of MFB in forecasting future returns. Additionally, discussing the robustness of the findings and potential confounding factors would enhance the credibility of the methodology.

R: In fact, both Section 4 (Cross-sectional return patterns associated with MFB) and Section 5 (Robustness tests) of our paper examine whether MFB is able to predict future stock returns using cross-sectional analysis. The results of univariate portfolio analysis demonstrate that MFB can predict stock returns, while bivariate portfolio analysis and Fama-Macbeth regression include potential confounding factors (such as other firm-specific characteristics: Beta, Size, BM, STR, MoM, Illiq, Coskew, BD, VOLDU, and VaR; multiple pricing factors: FF5+SL2+UMD+SENT) to evaluate the predictive power of MFB. The results suggest that even after controlling for these potential confounding factors, MFB is still able to predict future stock returns, which is also supported by out-of-sample tests. However, the stability test of the original manuscript may be weakened. In the revised manuscript, we provide further analysis by taking into account the impact of macroeconomic factors (subsample analysis), the discussion of risk perspective, and a reference point (a threshold for market up or down).

5. Comparison with Previous Studies: The authors briefly mention Levi and Welch (2020) as a previous study, but more discussion and comparison with relevant literature is needed. How does the proposed methodology differ from and build upon existing approaches to measuring framing effects and behavioural biases in financial markets? Highlighting the unique contributions and advantages of the current methodology would strengthen the manuscript's originality.

R: Thank you very much for the reviewer's comments. In the revised manuscript, we emphasize the comparative analysis of our proposed MFB with existing literature. On Introduction Page 3, we have added the following description:

The papers by Ang et al. (2006) and Levi and Welch (2020) explore the impact of downside beta on future returns from the perspective of risk (market risk exposures are time-vary). This paper, on the other hand, is the first to explore the significance of the difference between upside and downside beta from the perspective of investor psychology and behavioral biases, which is the biggest innovation of this paper. For example, Delta Beta is defined as the difference between downside beta and upside beta (Levi and Welch, 2020), which can be understood as the difference between downside exposure and upside exposure. But if you take an absolute value for Delta Beta, it's hard to explain from a risk perspective. However, the definition of frame bias is as long as there is "deviation", it does not involve "direction". Therefore, the absolute value of Delta Beta is interpreted as the bias of the risk-return trade-off between the up and down markets (MFB), which is a better operation than the risk perspective and is how we will calculate MFB in the next section.

Furthermore, we add references 3, 16 and 33. And in order to compare with existing literature, we have made additional efforts to the stability tests in the revised manuscript. We include a comparison of the difference between the down/up beta with and without taking the absolute value, as well as an impact analysis of different reference points. For detailed analysis and results, please refer to the stability tests section in the revised manuscript.

In summary, while the manuscript presents a methodology for quantifying MFB, there are several areas that require further clarification, empirical evidence, and comparative analysis. Addressing these points would improve the overall strength and reliability of the methodology section.

1. Methodological Details: The manuscript lacks detailed explanations of the specific calculations and data sources used to measure MFB and construct the decile portfolios. Providing step-by-step explanations of the calculations, including the formulas used and the specific databases or sources of data, would enhance the replicability of the study.

R：1) In the second section of the revised draft, "Quantification of MFB," we provide a detailed account of the origin of the MFB definition. Formula (1) and (2) provide a comprehensive definition of MFB.

Regarding the source of the data, we provide a description in the third section "Data and variables.": 

We collect the sample data for all A-shares (traded in the Shanghai Stock Exchange and Shenzhen Stock Exchange, excluding SSE STAR Market ). Daily and monthly stock market data used in this paper are from the RESSET database, except for the following data that come from China Stock Market & Accounting Research Database (CSMAR): momentum (UMD; Carhart, 1997), sentiment (SENT; Baker and Wurgler, 2006), monthly excess returns on the market (MKT), size (SMB), value (HML), investment (CMA) and profitability (RMW) factors of Fama and French (2015). The sample period is from January 2000 to December 2019, and stocks must have been traded for at least 36 months during the sample period. The final sample contains 3804 stocks and a total of nearly 500,000 firm-month observations.

Furthermore, the firm-specific characteristics are defined as follows. 1) Beta, following Bali et al. (2016), the market beta of each stock with respect to the value-weighted market excess return calculated from daily returns during the month. 2) Size, coming from Fama and French (1992), is calculated by the natural logarithm of each stock’s market capitalization at the end of each month. 3) BM, book-to-market equity ratio at the end of each month, which also comes from Fama and French (1992). 4) STR, a short-term reversal, derived from Jegadeesh (1990), is the return of a stock in the previous month. 5) MOM, the momentum return of each stock derived from Jegadeesh and Titman (1993) is the cumulative return during the past 11 months after skipping one month. 6) Illiq, illiquidity coming from Amihud (2002) is the absolute daily return divided by daily trading volume (hundred million yuan) averaged over all trading days in each month. 7) Coskew, the co-skewness shown by Harvey and Siddique (2000) is calculated as a daily regression coefficient for the model in each month. 8) BD, the downside beta shown by Ang et al. (2006) and Chiang (2019), is the sensitivity of each stock toward the excess market return during the days when the excess market return is below its mean during the month. 9) VOLDU, the difference between monthly money volume and its past 12-month average, which is derived from Atilgan et al. (2020). 10) VaR, value-at-risk also derived from Atilgan et al. (2020), is calculated as the 1st percentile of daily returns over the past 250 trading days at the end of the month.

2) The details about constructing the decile portfolios: 

A univariate portfolio analysis is performed where deciles are formed every month by ascending sorting stocks based on their MFB values. Excess returns and abnormal returns ( , adjusted by the FF5 factors) from 1 to 12 months ahead are calculated for each decile to test whether the zero-cost portfolio (denoted as portfolio H-L), which takes a long position in stocks with the highest MFB and a short position in stocks with the lowest MFB, has a significant return.

2. Interpretation of Results: While the summary table provides the excess returns and abnormal returns for each decile portfolio, the manuscript lacks a comprehensive interpretation and discussion of the findings. It is essential to provide a deeper analysis of the results, including their statistical significance, economic significance, and potential implications for investors. Additionally, connecting the findings back to the literature on the framing effect and behavioral finance would strengthen the manuscript's contribution.

R: Thank you for your suggestion. Whether it is univariate or bivariate analysis, for each decile portfolio, we provide excess returns and abnormal returns. If the future returns of the low decile combination are high or in other words, if portfolio H-L has significantly negative future returns, it suggests that the MFB used for grouping can predict future stock returns negatively.

In the revised draft, we strengthen the connection of the findings back to the literature. For example, Page 11, Lines 277-279: The results are consistent with Breuer and Soypak (2015), who emphasize that a stronger framing effect leads to a stronger status quo bias and negative outcomes. 

And in Lines 287-294, we make a comparative analysis: Our results differ from Ang et al. (2006)'s construction of Delta Beta (the difference between down beta and up beta) from the perspective of risk, for the following two main reasons: Firstly, we take the absolute value of Delta Beta from the perspective of investors' short-term behavioral biases, while it is hard to explain why Ang et al. (2006) take absolute values from the perspective of risk. Secondly, when calculating the upside/downside betas, we used daily data within each month based on investors' short-term behavioral biases, whereas Ang et al. (2006) used daily data from the past year.

Furthermore, we have conducted more detailed comparisons with additional references in the stability test of the revised draft, especially in the analysis of the "A risk perspective?" and "Reference point" sections. Your review on this matter would be greatly appreciated.

3. Bivariate Analysis: The manuscript briefly mentions a bivariate portfolio analysis but does not provide detailed results or discussion of this analysis. It would be valuable to present the findings of the bivariate analysis and explore the relationship between MFB and other firm-specific characteristics or behavior-related asset pricing factors. This would provide a more comprehensive understanding of the factors that influence MFB and their impact on stock returns.

R: Since our bivariate combination analysis involves 10 company characteristics, future returns for the next 1, 3, 6, and 12 months, as well as two different types of pricing factors, and also deals with the distinction between excess return and abnormal return, listing all the specific combinations for the groups would require a 240 (4×3×10×2) rows and 11 columns (including H-L combinations) table, which would be too large. Therefore, we mainly provide information on the H-L combinations. However, in Table 7, we do specify the details of each combination when dividing the sub-samples into economic booms and recessions according to macroeconomic factors. The results of our bivariate combinations are consistent with the data performance of the sub-sample and the subsequent Firm-level Fama-MacBeth regressions. Specifically, we find a negative correlation between MFB and future stock returns that is unaffected by other 10 company characteristics, indicating that MFB has predictive power.

4. Limitations and Robustness: The manuscript does not address potential limitations of the methodology or conduct robustness tests to assess the stability and reliability of the findings. It is crucial to acknowledge any limitations of the study, such as data limitations or assumptions made, and discuss how these limitations may affect the interpretation of the results. Additionally, conducting sensitivity analyses or alternative model specifications would enhance the robustness of the findings.

R: Thank you very much for your suggestions. Regarding the potential limitations of our methodology, we have added three sections for robustness analysis in the revised draft. In the "Subsample analysis based on the boom and recession" section, we have taken into account the impact of macroeconomics and analyzed the issue that stocks tend to exhibit stronger co-movement during market downturns (Das et al., 2018). In the "A risk perspective?" section, we have presented the consequences of not taking the absolute value, demonstrated the shortcomings from a risk perspective, and further proved the importance of analyzing from the perspective of short-term behavioral bias of investors. In the "Reference point" section, we have addressed the inadequacies of our reference point selection and presented the enhancement of the robustness of our findings by selecting different reference points.

---

## [Editor Report · Decision Letter 2]

10 Aug 2023

Market framing bias and cross-sectional stock returns

PONE-D-23-05212R2

Dear Dr. Gao,

We’re pleased to inform you that your manuscript has been judged scientifically suitable for publication and will be formally accepted for publication once it meets all outstanding technical requirements.

Kind regards,

Bartosz Gebka, PhD

Academic Editor

PLOS ONE

---

## [Editor Report · Acceptance letter]

18 Aug 2023

PONE-D-23-05212R2 

Market framing bias and cross-sectional stock returns 

Dear Dr. Gao:

I'm pleased to inform you that your manuscript has been deemed suitable for publication in PLOS ONE. Congratulations! Your manuscript is now with our production department. 

Kind regards, 

on behalf of

Professor Bartosz Gebka 

Academic Editor

PLOS ONE